# MARS-FM: GENERATIVE MODELING OF MOLECULAR DYNAMICS VIA MARKOV STATE MODELS

**Kacper Kapuśniak**[1,*,†], **Cristian Gabellini**[2,3],
**Michael Bronstein**[1,4], **Prudencio Tossou**[2,3], **Francesco Di Giovanni**[2,3]
[1]University of Oxford, [2]Valence Labs, [3]Recursion, [4]AITHYRA

## ABSTRACT

Molecular Dynamics (MD) is a powerful computational microscope for probing protein function. However, the need for fine-grained integration and the long timescales of biomolecular events make MD computationally expensive. To address this, several generative models have been proposed to generate surrogate trajectories at lower cost. Yet, these models typically learn a fixed-lag transition density, causing the training signal to be dominated by frequent but uninformative transitions. We introduce a new class of generative models, **MSM Emulators**, which instead learn to sample transitions across discrete states defined by an underlying Markov State Model (MSM). We instantiate this class with MARKOV SPACE FLOW MATCHING (MARS-FM), whose sampling offers more than two orders of magnitude speedup compared to implicit- or explicit-solvent MD simulations. We benchmark MARS-FM's ability to reproduce MD statistics through structural observables such as RMSD, radius of gyration, and secondary structure content. Our evaluation spans protein domains (up to 500 residues) with significant *chemical* and *structural* diversity, including unfolding events, and enforces *strict sequence dissimilarity* between training and test sets to assess generalization. Across all metrics, MARS-FM outperforms existing methods, often by a substantial margin.

## 1 INTRODUCTION

Deep Learning has unlocked fast and accurate prediction of proteins' 3D structures (Jumper et al., 2021; Baek et al., 2021; Krishna et al., 2024; Abramson et al., 2024). However, these methods do not capture dynamic behavior of proteins (Lewis et al., 2024), whose conformational ensembles are governed by the Boltzmann distribution. To study such dynamics, the most reliable computational tool is **Molecular Dynamics (MD)** (Alder & Wainwright, 1959; Rahman, 1964; McCammon et al., 1977; Risken & Risken, 1996), which simulates atomic motion by integrating Newton's Law. *Long* MD trajectories provide samples from the Boltzmann distribution and reveal the mechanisms of biomolecular interactions, a key asset in drug discovery (De Vivo et al., 2016). Nevertheless, MD is costly, as events in biology occur over timescales vastly longer than the simulation timestep. This challenge has spurred a range of *enhanced sampling* methods to accelerate dynamics, often through non-physical forces (Laio & Parrinello, 2002; Hamelberg et al., 2004; Jiang & Roux, 2010; Sabbadin & Moro, 2014).

Recent works have proposed to avoid the computational burden of MD by using generative flows to sample from the Boltzmann distribution (Noé et al., 2019; Zheng et al., 2024; Klein & Noe, 2024). A key subclass is the family of **MD Emulators** (MD-Emus), which learn to *emulate* MD-sampling by modeling the transition density associated with a fixed *lag time* $\tau$ (Klein et al., 2023; Schreiner et al., 2023; Nam et al., 2024; Diez et al., 2024; Costa et al., 2024). That is, MD-Emus are models that generate future conformations $x(t + \tau)$ conditioned on some input frame $x(t)$. At inference, these methods are applied autoregressively to produce surrogate MD trajectories. Jing et al. (2024b) advanced this approach by training a model to generate multiple frames jointly, all separated by a fixed interval $\tau$. However, learning dynamics at a fixed lag time introduces key challenges. Short lag times limit the achievable speed-up, while long lag times can skip over important meta-stable states. More fundamentally, the training signal is dominated by frequent but uninformative transitions observed during the simulation, while high-barrier transitions that drive exploration remain underrepresented. As a result, MD-Emus may struggle to capture rare, large conformational changes (Figure 1, left).

---

*Work done as an intern at Valence Labs/Recursion

†Corresponding author: kacper.kapusniak@cs.ox.ac.uk
  Code is available at https://github.com/valence-labs/mars-fm

MD-Emus  MSM-Emus  MSM-Emus ⊕ MD-Emus

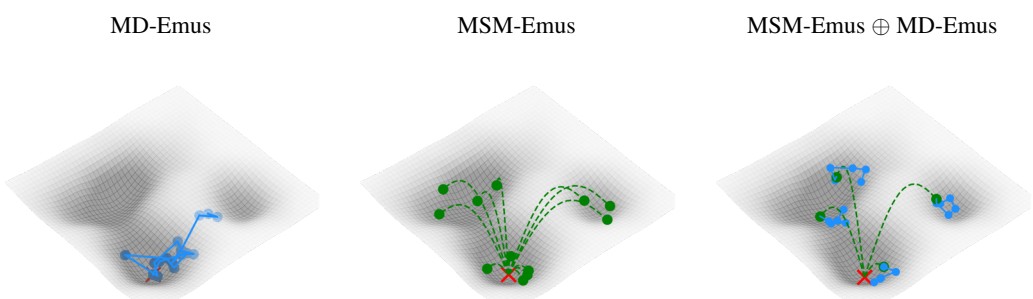

Figure 1: Comparison between existing approaches (MD-Emus) and our proposed novel class (MSM-Emus). MD-Emus learn transitions within a state (i.e. energy minima) well but could fail to generate transitions across different states (minima) since they are constrained by the data imbalance intrinsic to MD. Conversely, our framework learns to sample from the distribution induced by a Markov State Model (MSM). This modeling shift means that generative models are decoupled from temporal dynamics and can better learn to sample inter-state transitions. During sampling, MSM-Emus can generate conformations in parallel or be combined with existing MD-Emus to capture both large conformational changes as well as local dynamics within states.

Limitations of MD-Emus stem from the data imbalance in MD, whose temporal dynamic contains irrelevant high-frequency information. A common strategy to extract meaningful signal from MD through coarse-grained representations, is offered by **Markov State Models (MSMs)** (Noé et al., 2009; Prinz et al., 2011; Bowman et al., 2013). MSMs cluster frames into discrete *states* and describe dynamics via a Markov chain matrix $\mathsf{T}$. While compressing the dynamical content, MSMs guarantee estimation of long-time statistics through the equilibrium distribution induced by $\mathsf{T}$.

**Main Contributions**. We present **MSM Emulators (MSM-Emus)**, a new class of generative models that learn to sample from the Markov-chain transition induced by an MSM constructed from MD trajectories. In this view, the MSM provides a coarse-grained Markov model of long-timescale dynamics between metastable states, and MSM-Emus focus on learning transitions between metastable states rather than to the full high-frequency MD time series. This modeling shift has important implications as illustrated in Figure 1.

Since target transitions now depend on state connectivity rather than specific observed paths, MSM-Emus benefit from much higher training diversity for learning rare conformational changes across high-energy barriers. In fact, MSM-Emus interpolate across discrete macroscopic states associated with large conformational changes, such as folding or unfolding, rather than among specific frames, which is a more robust and generalizable signal across different proteins. During sampling, frames are more easily decorrelated and autoregressive calls are greatly reduced, mitigating compounding error effects. We showcase this class with MARKOV SPACE FLOW MATCHING (MARS-FM), a novel framework optimized using Flow Matching (Lipman et al., 2023).

**Evaluation with chemical and structural diversity**. MD-Emus have usually been evaluated on small peptides or datasets with limited chemical (Lindorff-Larsen et al., 2011; Majewski et al., 2023) or structural (Vander Meersche et al., 2024) diversity. To address this gap, we evaluate MARS-FM on MD-CATH (Mirarchi et al., 2024), a large-scale dataset of thousands of protein domains with sequence length up to 500 residues. In particular, by leveraging the highest-temperature replica, we are able to test MARS-FM's ability to capture large conformational changes, such as *unfolding*. To assess *generalization*, we adopt the protocol of Lewis et al. (2024) and use MMSEQS2 (Steinegger & Söding, 2017) with maximum sensitivity to ensure that **test proteins share no more than** 20% **sequence similarity with any protein in the training set**. We benchmark MARS-FM using structural observables reported in Jing et al. (2024b); Mirarchi et al. (2024), including RMSD, radius of gyration, and secondary structure content, and show that it consistently outperforms MD-Emus—often by a large margin. Crucially, MARS-FM explores the target distribution much **more efficiently than both MD and MD-Emus**, generating conformations across different states even in low-sample regimes.

## 2  PRELIMINARIES AND SETTING

A molecular system of $N$ atoms, in thermal equilibrium at temperature $T$, explores conformations according to the Boltzmann distribution $\mu$, i.e. $q \sim \mu(q) \propto \exp(-\mathcal{H}(q)/(k_B T))$ with $q = (x, \xi) \in \mathcal{X}$ denoting the system's position and velocity, $\mathcal{H}$ the Hamiltonian, and $k_B$ the Boltzmann constant. We

focus on proteins, whose *motions*—such as local unfolding or cryptic pocket formation—arise from structural fluctuations. To *identify* these conformational changes and *quantify* their frequencies (free energy), we aim to sample structures from the Boltzmann distribution $\mu$. This enables the measurement of **observables** $\phi$—functions defined on the phase space $\mathcal{X}$—via expectations $\mathbb{E}_{x \sim \mu}[\phi(x)]$.

Since direct sampling from $\mu$ is intractable, a widespread alternative is to use **Molecular Dynamics (MD)**. MD simulates a continuous-time Markov process, designed to be *ergodic* w.r.t. $\mu$. That is, for any observable, time averages over a long trajectory approximate expectations under $\mu$ (Schütte et al., 2023). A common formulation of MD is via **Langevin dynamic**: given a system with potential energy $\mathcal{U}$, position and velocity of an atom $i$ with mass $m_i$, are updated as follows

$$\dot{x}_i = \xi_i, \quad m_i \dot{\xi}_i = -\nabla_x \mathcal{U}(x) - \gamma m_i \xi_i + \sigma \dot{W}_t, \tag{1}$$

with $\gamma$ and $\sigma$ controlling friction and thermal noise. In practice, we discard the kinetic component and focus on the trajectories of positions $t \mapsto x(t)$. While MD is a powerful tool, its computational cost is a major bottleneck. Many biologically relevant events occur on millisecond timescales, while Langevin dynamics requires integration with time steps of the order of femtoseconds, making long simulations prohibitively expensive. Crucially, meta-stable states of a protein are often separated by high-energy barriers, leading to MD simulations wasting time trapped in deep local minima (Wales, 2005).

**Problem Formulation**. Given an initial protein conformation $x(0)$, MD produces an ensemble of structures by integrating eq. (1) and saving frames at regular intervals. Since MD is expensive, we aim to use generative models to sample surrogate distributions at a fraction of the cost. To enable generalization across proteins, a *single* model is trained using available MD trajectories from multiple sequences. At inference, we provide sequence and input conformation from an unseen protein, and the goal is to generate samples that are statistically indistinguishable from MD trajectories with respect to a set of physical observables $\phi$—that is, the model should match the distribution over structural or thermodynamic quantities that practitioners care about. Next, we review existing approaches that attempt to address this problem by sampling from a fixed transition density and discuss their limitations.

## 3 MD EMULATORS AND THE CHALLENGES OF FIXED LAG TIME TRANSITIONS

A key object describing MD sampling is the **transition density** $y \sim p_\tau(y|x)$ associated with a *lag time* $\tau$, which represents the probability of a state $x$ evolving to state $y$ within time $\tau$. Concretely, this is estimated from MD trajectories by examining future transitions $x(t + \tau)$ given an input frame $x(t)$. **MD Emulators (MD-Emus)** are a class of conditional generative models trained to approximate $p_\tau(\cdot|x(t))$ from MD data (Fu et al., 2023; Klein et al., 2023; Schreiner et al., 2023; Nam et al., 2024; Diez et al., 2024; Costa et al., 2024). Specifically, MD-Emus learn to generate future conformations $y \sim p_\tau^\theta(y|x(t))$ given $x(t)$. Common training approaches include Normalizing Flows (Rezende & Mohamed, 2015; Chen et al., 2018), Flow Matching (Lipman et al., 2023; Tong et al., 2023; Albergo et al., 2023), or Score Matching (Ho et al., 2020; Song et al., 2021).

These models train a neural network $v_\theta$—*conditioned* on sequence and an input conformation $x(t)$—to match the empirical distribution of transitions observed in the MD data. Jing et al. (2024b) refined such an approach in MDGen, by generating the next $K$ future conformations at once. Explicitly, MDGen learns to sample from the joint density $\mathbf{y} = (y_1, \ldots y_K) \sim p_{\tau,K}(\mathbf{y}|x) = \prod_i p_\tau(y_{i+1}|y_i)$, conditioned on $y_0 = x$. In the following, we assume we are given an MD-Emu that is trained to approximate $p_{\tau,K}$.

### 3.1 LIMITATIONS OF MD-EMUS

**Training**. One of the key challenges in learning from MD trajectories is the *data imbalance* between frequent but uninformative transitions in an energy minima, and rare but key transitions across different minima (Wales, 2005). To make things more concrete, consider the case where a protein MD trajectory visits two (or more) macroscopic states $S_A$ and $S_B$, such as folded and unfolded conformations, separated by high-energy barriers. Transitions $S_A \rightarrow S_B$ are rare but critical as they reveal conformational flexibility and functional dynamics. However in any given trajectory interval of length $K\tau$, the system is far more likely to remain in single state, than to cross an energy barrier. As a result, training batches for MD-Emus are typically dominated by high-frequency, low-information intra-state transitions. This data imbalance limits the model's ability to learn conditional distributions describing meaningful state changes. For example, to learn how to transition from folded to unfolded states, the model

must observe actual transitions $S_A \ni x(t) \to x(t + K\tau) \in S_B$ in the data—but such events are rare. This sparsity constrains the diversity and efficiency of training data and affects generalization, as MD-Emus learn to replicate fine-grained temporal resolution rather than large conformational changes.

**Inference**. MD-Emus need to be used autoregressively to generate *long* trajectories. This may lead to error accumulation, causing samples to progressively drift away from the data manifold. Nam et al. (2024) introduced a refinement step to address this, yet this comes at the expense of training a second network. Besides, using a larger $K$ does not fully resolve the problem, as the number of required samples often exceeds the maximum window feasible during training for larger proteins.

The challenges faced by MD-Emus stem from fixed lag time transitions often being uninformative and failing to capture meaningful conformational changes across different energy minima. To overcome these limitations, we first need to apply a coarse-grained representation to MD data so as to discard high-frequency information. To this aim, we rely on Markov State Models (MSMs) and introduce a novel class of generative flows to learn transitions directly among states (rather than frames). The key advantage is that, once trajectories have been clustered into metastable states, we are no longer restricted to training only on frame pairs of the form $x_t \to x_{t+\tau}$ obtained by slicing the time series at a fixed resolution. Instead, any pair of frames assigned to successive MSM states can be used for training, greatly increasing the number and diversity of informative transitions.

## 4 THE NEW CLASS OF MSM EMULATORS

**Markov State Models** (MSMs) provide a coarse-grained representation that removes high-frequency time signal while still capturing long-timescale statistics (Noé et al., 2009; Pande et al., 2010; Prinz et al., 2011; Husic & Pande, 2018). In MSMs, frames $x(t)$ are assigned to discrete states $S_1, \ldots, S_M$ and the dynamics is modeled as a Markov chain over these states. Given an interval $\tau$, one estimates a **transition matrix** $\mathsf{T}$, where $\mathsf{T}_{ij}$ is the probability of transitioning from $S_i$ to $S_j$ within time $\tau$:

$$\mathsf{T}_{ij} = \frac{C_{ij}}{\sum_k C_{ik}}, \quad C_{ij} = |\{x(t) \in S_i : x(t + \tau) \in S_j\}|. \tag{2}$$

Explicitly, that means we consider a probability density $p_{\mathsf{T}}(\cdot|x(t))$ satisfying

$$\int_{S_j} p_{\mathsf{T}}(y|x(t))dy = \mathsf{T}_{ij}, \quad \forall x(t) \in S_i, \ i, j \in \{1, \ldots, M\}. \tag{3}$$

In its simplest form, MSMs assumes a uniform density within each state. That is, given $x(t) \in S_i$ and $y \in S_j$, $p_{\mathsf{T}}(y|x(t))$ depends only on the identity of the clusters, but not on the specific conformations. *This coarse-graining enables robust estimation of long-timescale dynamics, even from limited transition data*. In principle, one could initialize *short* MD simulations (of length $\tau$) across discrete states $\{S_i\}$, and propagate their dynamics via $\mathsf{T}$. However, sampling from the states of an underlying (unseen) MSM remains a challenge. We propose a class of generative models that addresses this issue.

**Definition 1  MSM Emulators** *(**MSM-Emus**) are generative models that are trained to sample conformations from $p_{\mathsf{T}}(\cdot|x(t)))$, where $\mathsf{T}$ is the Markov chain transition matrix of a given MSM.*

For a frame $x(t) \in S_i$, the MSM defines a categorical distribution over successor states $j \mapsto \mathsf{T}_{ij}$. We interpret $p_{\mathsf{T}}(\cdot \mid x(t))$ as the resulting mixture distribution over MD frames: one first samples a destination state $S_j$ according to $\mathsf{T}_{ij}$ and then samples a conformation from the empirical ensemble of MD frames assigned to $S_j$. MSM-Emus are thus trained to draw conformations from this MSM-induced mixture, rather than to learn the exact frame-to-frame MD transition density. By matching the distribution $p_{\mathsf{T}}(\cdot|x(t))$—that depends only on the MSM states—MSM-Emus bypass many of the data scarcity and imbalance issues that affect MD-Emus §3.

**Increased sample diversity**. MSM-Emus *decouple* the learning objective from the specific transitions observed in the simulation. Rather than learning frame-to-frame transitions $x(t) \to x(t + \tau)$, they learn *state-to-state transitions* $S_i \to S_j$, enabling broader generalization (as validated in §5). Unlike MD-Emus, which attempt to replicate the exact temporal dynamics at fixed lag time $\tau$, MSM-Emus focus on capturing *macroscopic* transitions identified by the MSM. In the scenario described in §3, MSM-Emus learn to generate transitions $S_A \ni x \to y \in S_B$ from diverse starting points $x \in S_A$ beyond those explicitly visited by the MD simulation. Crucially, the kinetics defined

by the underlying MSM remains intact: during training, transitions are sampled from the distribution $\mathsf{T}$, ensuring that the global dynamical behavior is faithfully preserved.

**Fast exploration**. At inference, MSM-Emus can explore the energy landscape by generating samples $y \sim p_\mathsf{T}(y|x(0))$, without being bound by fine-grained temporal dynamics. This significantly improves *sampling efficiency* (as visualized in Figure 1 and validated in §5). Additionally, as many conformations can now be generated in *parallel*, MSM-Emus reduce compounding errors due to autoregressive calls. While MSM-Emus are designed to preserve the coarse-grained kinetics encoded by the MSM (e.g., metastable-state populations and transition probabilities), they are not intended to faithfully reproduce all fine-grained dynamical observables of the MD trajectories, at the same level of accuracy as fixed-lag MD-Emus. By construction, training on state-to-state jump transitions prioritizes long-timescale behavior over the detailed intra-state time ordering. In this sense, MSM-Emus are complementary to MD-Emus: the former emphasize efficient exploration and long-timescale thermodynamics/kinetics, whereas the latter more directly track the local, frame-to-frame dynamics (see Appendix C.2 for a concrete hybrid instantiation and torsion-angle decorrelation analysis).

### 4.1 A REPRESENTATIVE FRAMEWORK: MARKOV SPACE FLOW MATCHING

We introduce MARKOV SPACE FLOW MATCHING (MARS-FM), a novel representative framework of the MSM-Emu class. First, we describe concrete MSM instantiations defined over MD trajectories.

**Constructing MSMs**. Specific versions of MARS-FM are instantiated by first building an MSM. To this aim, we use standard tools (Scherer et al., 2015; Hoffmann et al., 2021). The states $\{S_i\}$ are defined in a lower-dimensional space of collective variables. A common approach for dimensionality reduction is *Time-lagged Independent Component Analysis* (TICA) (Pérez-Hernández et al., 2013; Wu et al., 2017), which identifies directions $\mathbf{w}_j$ that maximize the autocorrelation of $\mathbf{w}_j^\top x(t)$ at lag time $\tau$. Alternatively, clustering can be applied directly to observables such as radius of gyration and fraction of secondary structures (Figure 2). Given states, the Markov chain transition matrix $\mathsf{T}$ is estimated at a lag time $\tau$.

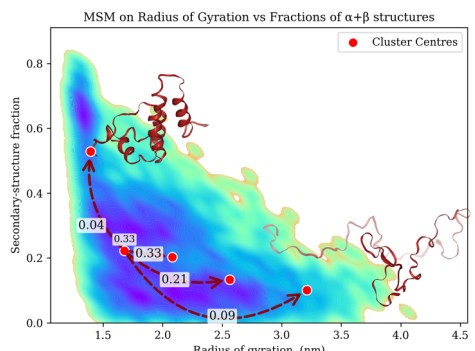

Figure 2: Clustering of MD conformations of protein 3ma5A00 from MD-Cath. We report cluster centres and Markov chain transitions from one representative state. We note how MSM states capture large structural differences (folded vs unfolded).

Two key observations are worth emphasizing. First, MSM construction is carried out once as an offline pre-processing step: for each dataset we fix the MSM hyperparameters and then build a separate MSM for every training domain from its MD trajectories. These per-domain state spaces and transition matrices remain fixed throughout training, ensuring minimal computational overhead. Second, since MARS-FM learns to interpolate between MSM states beyond the transitions directly observed in MD, we can afford to select a larger $\tau$ than MD-Emus without sacrificing the data availability. The lag time still sets the coarse-grained timescale of the underlying Markov chain, but the number and diversity of effective training transitions is no longer tied to how often frames are saved in the MD trajectories: any pair of frames assigned to successive MSM states can be used to define a training pair.

We emphasize that MSMs are defined as pre-processing of training data; importantly, **no MSM information is provided during inference**. Additional details on how we construct MSMs are reported in Appendix B.

**Representation**. To generate protein conformations, MARS-FM adopts the same SE(3) representation as (Jumper et al., 2021; Jing et al., 2024b). Each residue $\ell$ of frame $x(t)$ is represented as

$$\chi^\ell(t) = (q^\ell(t), r^\ell(t), (\cos(\theta_k^\ell(t)), \sin(\theta_k^\ell(t))_{k=1}^7), \tag{4}$$

where $q^\ell(t) \in \mathbb{R}^4$ is a unit quaternion describing a rotation and $r^\ell(t) \in \mathbb{R}^3$ is a translation, while $\theta_k^\ell(t)$ represent the $k$ torsion angles. Given an input $x(t)$, the target conformation $y$ is represented as an offset in roto-translational space. To model a vector field $v_\theta$ transporting a source distribution to the

target distribution $p_T(\cdot|x(t))$, we build on the MDGen architecture (Jing et al., 2024b). Specifically, $v_\theta$ leverages DiT-style blocks (Peebles & Xie, 2023) and is *conditioned* on protein sequence $a$ and the current conformation $x(t)$ via IPA layers (Jumper et al., 2021)—see Appendix A for a complete description. For notational simplicity, we omit the explicit conditioning on the protein sequence $a$.

**Training objective**. We train MARS-FM using **Flow Matching**, which learns a time-dependent vector field $v_\theta$ to transport a source distribution $p_0 = \mathcal{N}(0,1)$ to the target distribution $p_1(\cdot|x(t)) = p_T(\cdot|x(t))$. Concretely, given an input frame $x(t) \in S_i$, we first draw a destination state $S_j$ according to the MSM transition probabilities $j \mapsto T_{ij}$ and then choose a target MD frame among those assigned to $S_j$. In practice we sample *uniformly from the finite set of MD frames in* $S_j$. We denote the resulting target frame by $x_1$.

Independently, we draw a noise sample $x_0 \sim p_0$ and define interpolations $[0,1] \ni s \mapsto x_s$ connecting $x_0$ to $x_1$ using the standard schedule of Flow Matching (see Appendix A.2 for the explicit formula for $x_s$ and its analytic velocity $\dot{x}_s$). We optimize $v_\theta$ by minimizing the velocity mismatch along the conditional paths $s \mapsto p_s(\cdot \mid x_0, x_1)$ induced by these interpolations (Lipman et al., 2023; Liu et al., 2022; Albergo & Vanden-Eijnden, 2022). That is, we consider the training loss

$$\mathcal{L}_{\text{MARS-FM}}(\theta) = \mathbb{E}_{i \sim [M], x(t) \sim S_i, s \sim [0,1], x_0 \sim p_0(x_0), x_1 \sim p_T(x_1|x(t))} \|v_\theta(s, x_s; x(t)) - \dot{x}_s\|^2, \quad (5)$$

where $\dot{x}_s$ is the velocity of the conditional path. Crucially, we first select each state $S_i$ uniformly (see Appendix A.1) before conditioning on a specific frame $x(t) \in S_i$. This state-based sampling ensures that rare states are encountered more frequently during training. In contrast, standard MD-Emus draw frames uniformly from the trajectory, which often biases training toward common intra-state transitions. A similar strategy was used by Costa et al. (2024), though their model still approximates a fixed lag time transition. Further implementation and batching details are provided in Appendix A.

Compared to MDGen, which always takes $x_1$ to be a future frame $x(t + K\tau)$ from the same trajectory, MARS-FM samples $x_1$ via the MSM transition kernel $T$, so that the same Flow Matching objective is applied to a more diverse, state-conditioned set of training pairs. In particular, the model is not trained solely on transitions of the form $x(t + K\tau)$: the MSM provides transitions between all pairs of frames assigned to successive metastable states. Because each metastable state aggregates frames from different replicas and from multiple visits to the same basin within a trajectory, this construction exposes MARS-FM to transitions that span replicas and multiple barrier-crossing events, even if only a few such events occur in any individual time series. The lag time used to define the MSM still determines the coarse-grained timescale of these moves, but the effective training set is no longer restricted to the specific $x(t) \to x(t + \tau)$ pairs obtained at a fixed frame-saving interval. While these transitions still implicitly correspond to a characteristic timescale set by the MSM lag time, the number and diversity of transitions available for training are no longer constrained by the temporal resolution used to slice the trajectories.

**Sampling**. Following standard evaluation for MD-Emus, we assume that at inference, we have sequence $a$ and an input conformation $x(0)$ from an unseen protein. We explore two sampling strategies. *2. MARS-FM ⊕ MD-Emu. 1. Tree Sampling*. We apply MARS-FM following a hierarchical sampling scheme (Figure 3). We first generate $n$ frames $\{y_1, \ldots, y_n\} \sim p_T(\cdot|x(0))$ in *parallel*. Next, we generate $n$ frames from $p_T(\cdot|y_i)$ for each $y_i$. We continue extending the depth of the tree-sampling scheme based on our sample budget. We combine MARS-FM with MDGen (Jing et al., 2024b). We first sample conformations using MARS-FM and then use MDGen to generate shorter trajectories from each of these points, separately. This approach *mirrors the MSM paradigm*, where statistics are inferred by initiating short simulations from different states. We introduce such a hybrid scheme, as in certain workflows temporal fidelity may also be relevant (De Vivo et al., 2016).

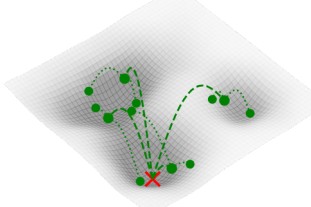

Figure 3: Hierarchical sampling used for MARS-FM.

For both cases, most conformations can be generated in parallel hence reducing autoregressive sampling. Crucially, as samples of MARS-FM are decoupled from temporal dynamics, they can explore the target distribution more efficiently (see Figure 5).

## 5 EXPERIMENTS

A key condition for a generative model to replace MD-sampling, is that measurements of observables under the generated samples match those under MD. For this reason, in our experiments, we compare distribution of observables computed along MD trajectories and those computed along conformations generated by the underlying model. We rely on observables used in previous generative works (Jing et al., 2024a;b) as well as additional ones reported for analyzing the MD-Cath dataset (Mirarchi et al., 2024).

**Variants of MARS-FM**. We first construct per-domain MSMs with dataset-specific hyperparameters—and then train MARS-FM to sample from $p_T(\cdot|x(t))$. Yet *no information about MSMs is provided during inference*. As explained in §4, during sampling, we consider two variants: MARS-FM $\oplus$ MDGen and MARS-FM.

**Baselines**. In our experiments, we take MDGen (Jing et al., 2024b) as the main representative of the

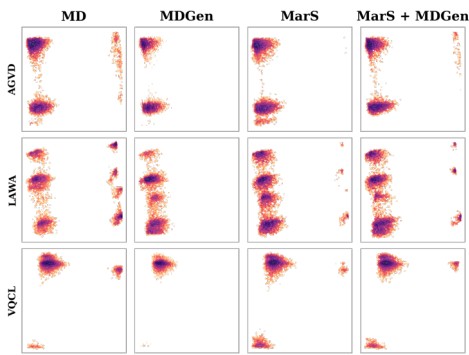

Figure 4: TICA plot for 3 random peptides in the test set, comparing MD ground-truth, MDGen, MARS (ours) and MARS + MDGen (ours). Our frameworks explore modes that are otherwise entirely ignored by MDGen. Similar plots are reported in Appendix C.

MD-Emu class. Primarily, this is due to MDGen being the most versatile variant of MD-Emus due to the choice of the window $K$ of consecutive transitions at resolution $\tau$. For larger systems, we additionally compare against BioEmu (Lewis et al., 2024), noting specific caveats discussed in Appendix B.7. Finally, we point out that other methods are either *not* scalable for MD-Cath domain (Klein et al., 2023; Schreiner et al., 2023) or have only been validated within the training distribution (Costa et al., 2024) (see Appendix B.7 for further discussion).

### 5.1 PRELIMINARY INVESTIGATION: TETRAPEPTIDES

We first adopt the dataset of tetrapeptides that provided the main evaluation for MDGen (Jing et al., 2024b). This consists of ~3000 trajectories in training, 100 in validation and 100 in test. Each trajectory is simulated in explicit solvent for 100 ns and conformations are saved every 10 ps. That is, for any tetrapeptide in the test set, we have a held-out MD distribution of $10^4$ frames. Accordingly, we generate distributions of $10^4$ frames using both MD-Emus (MDGen) and MSM-Emus (MARS-FM).

**MSM construction**. When training MARS-FM, we need to define MSMs over the MD trajectories. As this dataset contains exhaustive simulations relative to the size of the systems, we rely on TICA (Pérez-Hernández et al., 2013), selecting the minimal number of TICA coordinates whose cumulative kinetic variance exceeds 95%. We then apply k-means clustering to these coordinates, obtaining 100 microstates, which we further group into 10 metastable states using the PCCA+ spectral clustering method (Röblitz & Weber, 2013). This procedure yields the final MSMs constructed with a lag time of 100 ps. Further details on MSM construction are provided in Appendix B.

**Evaluation and metrics**. During sampling, we report the Jensen-Shannon Divergence (JSD) over different observables. Namely, we follow the same metrics as in MDGen Jing et al. (2024b): (i) Torsional angles (of both backbone and sidechains); (ii) Projection onto the slowest modes as identified by TICA; (iii) Equilibrium distributions induced by MSMs built over the test peptides. Inspired by Lewis et al. (2024), we also report the Macrostate MAE (mMAE), which measures the mean absolute error between model-generated and ground-truth MD free energies across metastable

Table 1: JSD ↓ between sampled and ground-truth distributions for Tetrapeptides. Results based on sampled trajectories of $10^4$ conformations compared to ground-truth distributions.

|  | Torsions (bb) | Torsions (sc) | Torsions (all) | TICA-0 | TICA-0,1 joint | MSM states | Macrostate MAE |
|---|---|---|---|---|---|---|---|
| MD (Oracle) | 0.10 | 0.06 | 0.08 | 0.20 | 0.27 | 0.21 | — |
| MDGen-1000 | 0.13 | **0.09** | 0.11 | 0.23 | 0.32 | 0.23 | 1.13 |
| MDGen-200 | 0.14 | 0.10 | 0.12 | 0.24 | 0.33 | 0.27 | 1.12 |
| MARS-FM $\oplus$ MDGen-200 (ours) | 0.12 | **0.09** | **0.10** | **0.21** | 0.30 | 0.23 | 0.83 |
| MARS-FM (ours) | **0.11** | **0.09** | **0.10** | **0.21** | **0.29** | **0.22** | **0.63** |

Table 2: Pearson $r$ ↑ for RMSD and RMSF, forward KL divergence ↓, JSD ↓ of gyration radius, fraction of secondary structures, and MSMs distribution, and folding free energy MAE (kcal/mol) ↓. Results based on sampled trajectories of 500 conformations compared to ground-truth distributions and averaged over 5 inference runs.

| | Pairwise RMSD r | Global RMSF r | Per target RMSF r | Gyration Radius KL | Gyration Radius JSD | Secondary Structures KL | Secondary Structures JSD | MSM JSD | $\Delta G_{\text{fold}}$ MAE |
|---|---|---|---|---|---|---|---|---|---|
| MD (Oracle) | 0.82 ± .014 | 0.82 ± .012 | 0.90 ± .005 | 0.58 ± .020 | 0.07 ± .001 | 0.61 ± .048 | 0.07 ± .003 | 0.19 ± .006 | 0.90 ± .046 |
| MDGen-100 | 0.42 ± .047 | 0.49 ± .018 | 0.64 ± .007 | 1.13 ± .025 | 0.15 ± .003 | 1.17 ± .045 | 0.18 ± .005 | 0.29 ± .005 | 1.21 ± .010 |
| MDGen-20 | 0.40 ± .016 | 0.38 ± .013 | 0.38 ± .013 | 0.89 ± .010 | 0.20 ± .002 | 1.85 ± .024 | 0.30 ± .002 | 0.43 ± .009 | 1.44 ± .026 |
| MDGen-100 (in parallel) | 0.42 ± .011 | 0.54 ± .003 | 0.69 ± .009 | 2.15 ± .019 | 0.22 ± .001 | 1.83 ± .015 | 0.20 ± .002 | 0.39 ± .014 | 2.14 ± .015 |
| MDGen-20 (in parallel) | 0.43 ± .002 | 0.55 ± .001 | 0.70 ± .005 | 2.71 ± .005 | 0.26 ± .001 | 2.96 ± .008 | 0.28 ± .001 | 0.46 ± .002 | 3.68 ± .007 |
| BioEmu | 0.25 ± .002 | 0.41 ± .004 | 0.66 ± .001 | 3.83 ± .011 | 0.40 ± .001 | 4.17 ± .015 | 0.41 ± .001 | 0.51 ± .001 | 4.67 ± .004 |
| MARS-FM ⊕ MDGen-20 | 0.63 ± .004 | 0.69 ± .002 | 0.83 ± .001 | 0.98 ± .017 | 0.13 ± .002 | 0.73 ± .005 | 0.11 ± .001 | 0.24 ± .002 | 1.02 ± .002 |
| MARS-FM | 0.65 ± .004 | 0.71 ± .003 | 0.89 ± .001 | 0.55 ± .002 | 0.10 ± .001 | 0.93 ± .010 | 0.13 ± .001 | 0.19 ± .004 | 1.05 ± .003 |

states. The mMAE provides a direct measure of the model's accuracy in reproducing metastable state distributions. Further computational details and definitions are presented in Appendix B.

**Results**. For the baselines, we train MDGen with window size $K = 200$ and $K = 1000$, and use the former combined with MARS-FM as per our hybrid-scheme described in §4. Results in Table 1 illustrate that MARS-FM achieves same or better performance than baselines. In particular, errors are often comparable with those encountered by different replicas of MD trajectories (oracle performance). Noticeably, a large improvement is achieved on the Macrostate MAE, highlighting that MARS-FM can better sample from rare meta-stable states of the underlying TICA space of unseen peptides. This is also confirmed in Figure 4—additional TICA plots are reported in Appendix C. In general though, we expect MD-Emus and MSM-Emus to be comparable for this dataset, as we have very limited chemical diversity and no large domain motion due to the size of the systems.

## 5.2 MD-CATH: PROTEINS WITH SEQUENCE DISSIMILARITY AND LARGE DOMAIN MOTIONS

Our main evaluation is based on the MD-Cath dataset from Mirarchi et al. (2024). This includes 5398 domains up to 500 residues (*average number of residues* 137), simulated at 5 different temperatures, spanning 5 random replicas per setting. The average length of each trajectory is 464 ns and frames are saved at 1 ns intervals. The dataset is already built to ensure protein diversity by considering non-homologous domains at the $S20$ (20%) homology level—further details are reported in Appendix B.

**MSM construction**. MSMs are constructed independently for each domain using k-means clustering directly applied to normalized observables identified as critical by the original dataset analysis (Mirarchi et al., 2024). Specifically, we consider two features: (i) the radius of gyration and (ii) the fraction of residues in $\alpha$-helical and $\beta$-sheet secondary structures—additional details are reported in Appendix B. We cluster these normalized features into 10 metastable states, providing a physically meaningful partitioning of the conformational space. This yields the final MSMs constructed with a lag time of 50 ns.

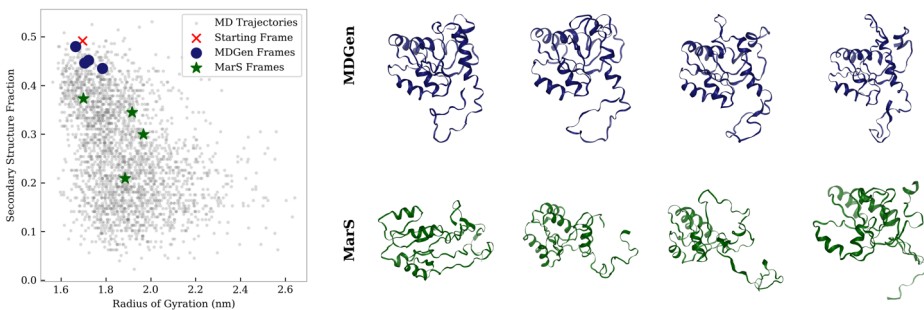

Figure 5: First 4 samples generated by MDGen and MARS-FM for the domain 2ynmD03 in the test set. As MARS-FM interpolates among states independently of temporal dynamics, it can explore the energy landscape more efficiently. In fact, the secondary structure content varies significantly among these 4 samples (note that there is no ordering as they are generated in parallel). Conversely, MDGen samples all belong to the same energy minimum which reduced sampling efficiency and exploration.

Table 3: Pearson $r$ ↑ for RMSD and RMSF, forward KL divergence ↓ and JSD ↓ of gyration radius, fraction of secondary structures, MSMs distribution, and folding free energy MAE (kcal/mol) ↓. Results based on sampled trajectories of 100 / 1000 conformations compared to ground-truth distributions and averaged over 5 inference runs.

| | Pairwise RMSD r | Global RMSF r | Per target RMSF r | Gyration Radius KL | Gyration Radius JSD | Secondary Structures KL | Secondary Structures JSD | MSM JSD | $\Delta G_{fold}$ MAE |
|---|---|---|---|---|---|---|---|---|---|
| MD (Oracle) | 0.65 / 0.89 | 0.70 / 0.87 | 0.77 / 0.92 | 2.19 / 0.32 | 0.18 / 0.05 | 2.75 / 0.26 | 0.22 / 0.05 | 0.49 / 0.12 | 2.40 / 0.80 |
| MDGen-100 | 0.34 / 0.28 | 0.46 / 0.32 | 0.60 / 0.46 | 3.66 / 0.78 | 0.31 / 0.14 | 3.60 / 1.49 | 0.29 / 0.26 | 0.51 / 0.27 | 2.58 / 1.52 |
| MDGen-20 | 0.57 / 0.28 | 0.62 / 0.23 | 0.61 / 0.20 | 2.48 / 1.37 | 0.22 / 0.33 | 2.08 / 2.34 | 0.19 / 0.39 | 0.48 / 0.51 | 1.52 / 2.01 |
| MDGen-100 (in parallel) | 0.37 / 0.43 | 0.46 / 0.56 | 0.62 / 0.71 | 3.62 / 1.84 | 0.31 / 0.21 | 3.51 / 1.54 | 0.28 / 0.19 | 0.52 / 0.38 | 2.68 / 2.12 |
| MDGen-20 (in parallel) | 0.41 / 0.43 | 0.53 / 0.55 | 0.67 / 0.71 | 3.56 / 2.50 | 0.30 / 0.26 | 3.87 / 2.75 | 0.31 / 0.27 | 0.53 / 0.46 | 3.77 / 3.65 |
| BioEmu | 0.23 / 0.26 | 0.40 / 0.42 | 0.64 / 0.67 | 4.75 / 3.55 | 0.43 / 0.39 | 5.08 / 3.91 | 0.44 / 0.41 | 0.55 / 0.41 | 4.82 / 4.62 |
| MARS-FM ⊕ MDGen-20 | 0.59 / 0.64 | 0.66 / 0.71 | 0.79 / 0.84 | 1.99 / 0.77 | 0.20 / 0.12 | **1.85** / **0.61** | **0.18** / **0.11** | 0.43 / 0.23 | 1.38 / **1.02** |
| MARS-FM | **0.60** / **0.65** | **0.68** / **0.71** | **0.84** / **0.90** | **1.74** / **0.42** | **0.18** / **0.09** | 1.92 / 0.95 | **0.18** / 0.14 | **0.42** / **0.17** | **1.25** / 1.20 |

**Large domain motions**. To validate the ability of MARS-FM to generate large conformational changes, our main evaluation is performed over the highest-temperature replica (450 K), in which proteins exhibit ***unfolding***. In fact, proteins are quite stable at lower temperatures over the simulated timescales, meaning that often no significant conformational change has occurred—as reported in Mirarchi et al. (2024)—which would result in an easier generative task. Nonetheless, for a more complete evaluation, we also train MARS-FM on lowest-temperature replica and sample conformations in this regime–results are reported in Appendix C.

**Assessing generalization**. First, we consider a random training-validation-test split partitioning the total number of domains according to 80%-10%-10%. We then follow the strategy in Lewis et al. (2024) and use MMSEQS2 (Steinegger & Söding, 2017) with *highest sensitivity setting* to **filter out any test protein sharing more than** 20% **sequence similarity with any training protein** (note that Lewis et al. (2024) use a more forgiving 40% threshold). As a result, our test set consists of 495 domains that are meaningfully diverse from those whose trajectories have been used during training.

**Metrics and Observables**. For evaluation, we focus on observables studied in the original dataset analysis of Mirarchi et al. (2024), namely the radius of gyration and fractions of residues in secondary structures ($\alpha$-helices and $\beta$-sheets). We measure the forward KL divergence to quantify the model's exploration capabilities and the Jensen-Shannon Divergence (JSD) for general distribution alignment. Inspired by Jing et al. (2024a), we also quantify ensemble flexibility using the Pearson correlation ($r$) computed on pairwise backbone Root Mean Square Deviation (RMSD), global Root Mean Square Fluctuation (RMSF), and per-target RMSF. We additionally evaluate folding free energies via the mean absolute error ($\Delta G_{fold}$ MAE). Lastly, we calculate the JSD between reconstructed MSMs and reference MSMs, providing a direct measure of the ability to explore states of domains dissimilar from those observed in training. Additional details on metrics are provided in Appendix B.

**Extended baselines**. We benchmark different variants of MDGen to highlight how advantages of our framework *cannot be simply replicated by changing the lag time or altering the sampling approach*. As such, we train MDGen with window size $K = 20$ and $K = 100$, corresponding to a total lag time of 20 ns and 100 ns, respectively. Additionally, to demonstrate that the improvements of MARS-FM cannot be simply ascribed to a reduction of autoregressive calls, we also evaluate MDGen in *parallel*, meaning that we sample conformations only conditioned on the input frame. Finally, we also report the performance of one MD replica against 4 held-out ones (and average over all possible 5 combinations) to be treated as *oracle* performance.

**Results**. We report our evaluation in Table 2. Both hybrid sampling (MARS-FM ⊕ MDGen) and hierarchical sampling (MARS-FM) significantly improve over all extended MDGen baselines, often by a large margin. This confirms that our framework can extrapolate over unseen large conformational changes better than MD-Emus. Crucially, MARS-FM can even match the oracle performance in the reconstruction of the unseen underlying MSM. This validates how the framework is capable to generalize across unseen MSMs and points to such a signal being easier and more robust than the one derived from fixed-lag time transitions. As all test proteins share no more than 20 % sequence similarity to any training protein, we again emphasize that the results in Table 2 provide a stringent test for the generative models to sample conformations over meaningfully different domains.

**Ablation: Changing sample budgets**. Finally, we report results across different sample budget in Table 3. Namely, we consider the same metrics as above but compare distributions obtained by gener-

ating 100 and 1000 samples, respectively. In the low-sample regime, MARS-FM also **surpasses the oracle performance** as given by the first 100 frames (at 1 ns resolution) sampled by MD. This confirms that by decoupling the training objective from temporal dynamics, MARS-FM can sample large conformational changes much more efficiently as the model interpolates directly across states rather than across frames separated by a time interval. We illustrate this phenomenon in Figure 5. Our framework also offers optimal performances at higher sapling regimes, as it significantly reduces autoregressive calls hence mitigating compounding error effects. Overall, the results in Table 3 confirm that MARS-FM improves upon the baselines across different sample budgets due to (i) Its ability to sample large-domain transitions independent of temporal ordering; (ii) Reduced autoregressive sampling.

**Sampling speed**. A key requirement for generative models to serve as practical replacements for MD is that they offer substantial computational speed-ups. Table 4 summarizes the wall-clock time required to generate the equivalent of 500 ns of trajectory (500 conformations) for a 159-residue protein (domain 4dhkB00) on an NVIDIA H100. For a fair comparison, we report MD in *implicit solvent*, which is substantially more efficient than explicit-solvent simulations. For such a system, implicit MD runs at $\approx 2400$ ns/day on the same hardware for a 159-residue protein (OpenMM Team, 2025). Under these conditions, MARS-FM provides a computational speed-up of $600\times$.

Table 4: Wall-clock time (in seconds, ↓) to generate the equivalent of 500 ns trajectory (500 conformations) for a 159-residue protein.

| Method | Time (s) ↓ |
|---|---|
| MD (implicit solvent) | $\approx 18,000 \ (= 5\text{h})$ |
| MDGen-100 | $31.70 \pm 0.21$ |
| MDGen-20 | $100.87 \pm 2.24$ |
| MARS-FM $\oplus$ MDGen-20 | $19.79 \pm 0.04$ |
| MARS-FM | $30.34 \pm 0.08$ |

## 6 CONCLUSIONS

**Related works**. Beyond MD-Emus, our work is related to the class of Boltzmann Generators (BGs), normalizing flows that are trained via the potential energy (Noé et al., 2019; Wirnsberger et al., 2020; Köhler et al., 2021; Rizzi et al., 2021; Garcia Satorras et al., 2021; Rizzi et al., 2023; Midgley et al., 2023; Klein & Noe, 2024; Tan et al., 2025). However, BGs have shown limited scalability beyond small peptides. Alternatively, Jing et al. (2024a); Lewis et al. (2024) proposed sequence-to-structure generative flows trained on MD data, after ignoring any temporal ordering. These methods though suffer from the same data inbalance issues affecting MD-Emus. Learning to sample diverse protein conformations was also investigated in (Jing et al., 2023; Zheng et al., 2024; Lu et al., 2024) and is related to works that perturbed AlphaFold at the MSA level (Del Alamo et al., 2022; Wayment-Steele et al., 2024) or refined its predictions via experiments (Maddipatla et al., 2025). However, these works may fail to attain *quantitative* distributional matching. Finally, MSMs have been used, for a single peptide dataset, to reweigh samples when fine-tuning a generative model in Lewis et al. (2024).

**Limitations and Future Works**. In general, public MD data of protein dynamics is limited, particularly for long unbiased simulations. This generally hinders the development of generative models—a key reason as to why to assess large domain motions with sufficient chemical diversity we had to focus on higher-temperature simulations. In terms of scope, in this work we have focused on protein representation. Natural next steps would entail extending MSM-Emus to complexes, for example protein-ligand ones, and inorganic systems. MARS-FM requires an input 3D structure at inference. To further accelerate sampling and increase applicability of the framework, we will study how to leverage recent sequence-to-structure models to be able to generate protein conformations starting from sequence only. Finally, it would be interesting to explore whether MARS-FM can be combined with MD simulations, for example by sampling different input conformations across an underlying MSM, and then initializing shorter MD simulations in parallel.

ACKNOWLEDGMENTS

The authors would like to thank Daniel Cutting, Berton Earnshaw, Nikhil Shenoy, and Stephan Thaler for their helpful discussions, input, and feedback on this work. The authors also thank Lars Holdijk for his assistance with high-performance computing resources and support.

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

## Outline of Appendix

In Section A we describe the MARS-FM architecture, and our MSM-informed procedure for drawing training pairs; Section B reports dataset details, supplementary MSM construction choices, evaluation metrics, hyper-parameters, sampling settings, and compute resources; Section C gathers supplementary figures and tables—including tetrapeptide TICA plots, MD-Cath samples plots, expanded MD-Cath flexibility metrics, and the full 320K evaluation—that complement the main text; Section D summarizes usage of LLMs; finally, Section E reflects on the societal benefits and potential misuse risks of generative models such as MARS-FM and Section F contains reproducibility statement.

## A  Implementation Details

### A.1  Architecture

Inspired by MDGEN (Jing et al., 2024b), the velocity network $v_\theta : (\mathbb{R}^{21})^L \to (\mathbb{R}^{21})^L$, with $L$ being the overall number of residues, employs modified DiT blocks (Jing et al., 2024b; Peebles & Xie, 2023) and Invariant Point Attention (IPA) layers (Jumper et al., 2021). A pseudocode is given in Algorithm 1.

---

**Algorithm 1** Velocity Network

---

**Require:** conditioning tokens $\chi^{(0)} \in \mathbb{R}^{21L}$, target tokens $\chi_s^{(1)} \in \mathbb{R}^{21L}$, conditioning roto-translations $g^{(0)} \in SE(3)^L$, amino acid identities $a$, flow matching time $s \sim \mathcal{U}(0,1)$
**Ensure:** velocity $v \in \mathbb{R}^{21L}$
1: $s \leftarrow$ TIMEEMBED$(s)$
2: $x \leftarrow$ AMINOACIDEMBED$(a)$
3: **for** $\ell = 1$ **to** $n_{\text{IPA}}$ **do**
4:     $x \leftarrow$ INVARIANTPOINTATTENTIONLAYER$\big(x, g^{(0)}, s\big)$
5: $x \leftarrow x +$ LINEAR$(\chi^{(0)}) +$ LINEAR$(\chi_s^{(1)})$
6: **for** $\ell = 1$ **to** $n_{\text{DTA}}$ **do**
7:     $x \leftarrow$ DIFFUSIONTRANSFORMERATTENTIONLAYER$(x, s)$
8: **return** DIFFUSIONTRANSFORMERFINALLAYER$(x, s)$

---

At flow-matching time $s \in [0, 1]$, we construct the noisy representation as a linear combination of the ground-truth frame and i.i.d. Gaussian noise $\varepsilon$:

$$\chi_s^{(1)} = \sigma(s)\,\varepsilon + \alpha(s)\,\chi^{(1)}, \qquad \alpha(s) = \sin\big(\tfrac{\pi}{2}s\big), \quad \sigma(s) = \cos\big(\tfrac{\pi}{2}s\big).$$

The derivatives $\dot{\alpha}(s) = \frac{\pi}{2}\cos\big(\frac{\pi}{2}s\big)$ and $\dot{\sigma}(s) = -\frac{\pi}{2}\sin\big(\frac{\pi}{2}s\big)$ enter the target velocity $\dot{\chi}_s^{(1)}$ during training.

### A.2  Drawing training pairs

**Drawing Procedure.** Given a per-protein Markov State Model (MSM), for each training example we draw a pair $(x_0, x_1)$, as follows,

1. **Select source states.** We always include the state containing the first frame of the trajectory. The remaining source states are sampled uniformly (with replacement) from all Markov states.
2. **Draw destination states.** For each source state $S_i$, we sample a fixed number of destination states $S_j$ independently using the MSM transition probabilities $\mathsf{T}_{ij}$ as a categorical distribution.
3. **Select frames.** For each source–destination pair $(S_i, S_j)$, we uniformly sample one frame $x(t) \in S_i$ as the conditioning frame and one frame $x_1 \in S_j$ as the target frame. These form the training pair $(x_0, x_1)$. This step can be repeated a few times

This strategy balances sampling between rare and frequent metastable states while preserving the transition structure encoded in the MSM. Our sampling method allows to sample across different replicas.

## B   EXPERIMENTAL DETAILS

### B.1   DATASETS

**Tetrapeptides**   We use the tetrapeptides dataset introduced in Jing et al. (2024b), where trajectories are trained at a resolution of 10 ps over a total duration of 100 ns. We adopt the exact same split as introduced by the authors, comprising 3,109 tetrapeptides for training, 100 for validation, and 100 for testing.

**MD-Cath**   We use the MD-Cath dataset introduced in Mirarchi et al. (2024), which provides molecular dynamics simulations for 5,398 protein domains. Frames are trained at a resolution of 1 ns, with an average trajectory length of 464 ns per replica and the majority of trajectories being 500 ns long. Each domain (ranging from 50 to 500 residues, average 137) is simulated at five temperatures from 320 K to 450 K, with five replicas per temperature, yielding 25 trajectories per domain. For the train/validation/test split, we follow the similar strategy to Lewis et al. (2024) (but more strict) and use MMSEQS2 (Steinegger & Söding, 2017) with default parameters and maximum sensitivity to exclude from the test set any domain that shares more than 20% sequence identity with any training or validation domain. This results in a final split of 4,304 domains for training, 538 for validation, and 495 for testing. We primarily evaluate our models on the highest-temperature subset (450 K) of MD-Cath, where most proteins undergo (partial) unfolding within 100–500 ns. In contrast, lower temperature simulations (e.g., at 320 K) often remain near the native structure and exhibit limited conformational diversity—for a quantitative comparison see Appendix C. We use all available replicas for the given temperature.

### B.2   ADDITIONAL DETAILS ON MARKOV STATE MODEL CONSTRUCTION

We follow the MSM construction as described in Section 5. All MSMs are implemented using the `deeptime` library (Hoffmann et al., 2021). The resulting transition probability matrices are symmetrized post hoc as $\mathbf{P} \leftarrow (\mathbf{P} + \mathbf{P}^\top)/2$.

For clustering of MD-Cath dataset we use standardized Radius of Gyration and Secondary Structures Fractions as defined in the Appendix B.3.

### B.3   EVALUATIONS AND METRICS

#### B.3.1   TETRAPEPTIDES

**Evaluation metrics.**   We follow the evaluation protocol of Jing et al. (2024b) and compute the Jensen-Shannon Divergence (JSD) between model-generated and reference MD trajectories across several observables: (i) distributions torsional angles, (ii) TICAs, and (iii) equilibrium distributions derived from Markov State Models.

**Macrostate Mean Absolute Error (mMAE).**   To complement the evaluation and JSD-based observables, we additionally report the mMAE. The mMAE measures the discrepancy in metastable-state populations between the generated and reference ensembles, after both are coarse-grained into macrostates using the MSM calculated as described in Section 5. It is computed as the mean absolute difference in free energy across macrostates,

$$\text{mMAE} = \frac{1}{n}\sum_{i=1}^{n}\left|G_i^{\text{model}} - G_i^{\text{ref}}\right|, \quad \text{where} \quad G_i = -k_B T \log \pi_i.$$

Here, $G_i$ denotes the free energy of macrostate $i$, $k_B$ is the Boltzmann constant, $T$ is the temperature, and $\pi_i$ is the stationary distribution (normalized histogram count) of macrostate $i$. We use a small floor value (e.g., $10^{-4}$) to avoid numerical instability in the logarithm. Since we do not have access to additional replicas, we did not report the mMAE of the Oracle.

### B.3.2 MD-CATH

**Predicting Flexibility.** We follow the exact procedure of Jing et al. (2024a) to compute pairwise Root Mean Square Deviation (RMSD), global Root Mean Square Fluctuation (RMSF), and per-target RMSF. These metrics capture the flexibility and structural variability of conformational ensembles.

**Distribution Alignment Metrics.** To assess the similarity between model-generated and reference ensembles, we compute both the forward Kullback–Leibler (KL) divergence and the Jensen–Shannon Divergence (JSD). Both distributions are estimated by binning values into histograms with 100 bins and normalizing the counts to obtain discrete probability distributions. To avoid numerical issues, we apply a small floor value ($\epsilon = 10^{-5}$) to the model probabilities before computing KL. Specifically:

- The forward KL divergence, $D_{\text{KL}}(P \,\|\, Q)$, is computed with smoothed model probabilities: $Q = \max(Q, \epsilon)$.
- The Jensen–Shannon Divergence is computed as the squared JSD distance — `scipy`'s `jensenshannon` function (Virtanen et al., 2020).

**Radius of Gyration.** We compute the radius of gyration for each frame in a generated trajectory and compare its distribution against that of generated trajectories across all replicas at a given temperature. It is defined as,

$$R_g = \sqrt{\frac{\sum_{i=1}^{N} m_i \, \|\mathbf{x}_i - \mathbf{x}_{\text{center}}\|^2}{\sum_{i=1}^{N} m_i}}, \tag{6}$$

where $m_i$ and $\mathbf{x}_i$ denote the mass and position of atom $i$, and $\mathbf{x}_{\text{center}}$ is the center of mass of the structure. Higher $R_g$ values typically correspond to more extended or unfolded conformations.

**Secondary Structure Fractions.** We compute the fraction of residues in secondary structure per frame based on DSSP assignments. Specifically, we include canonical $\alpha$-helix and $\beta$-strand states, using `mdtraj`'s `compute_dssp` function (McGibbon et al., 2015),

$$f_{\text{SS}} = \frac{1}{L} \sum_{j=1}^{L} \mathbb{I} \left[ s_j \in \{\text{H, G, I, E, B}\} \right], \tag{7}$$

where $L$ is the number of residues in a frame, $s_j$ is the DSSP-assigned secondary structure of residue $j$, and $\mathbb{I}$ is the indicator function.

**Markov State Model Recovery.** We build MSMs using the same procedure as for training (Section 5), and then we compare the resulting stationary distributions via Jensen–Shannon Divergence.

**Folding Free Energies.** To estimate folding free energies, we follow the BioEmu protocol (Lewis et al., 2024). For each protein, we first compute a native–contact (FNC) score using all heavy–atom pairs with sequence separation $|i - j| > 3$ and reference distance $d_{ij}^{\text{ref}} < 10$ Å. For a given trajectory frame at time $t$, each pair contributes

$$q_{ij}(t) = \left[ 1 + \exp\left( -\beta \left[ d_{ij}(t) - \lambda d_{ij}^{\text{ref}} \right] \right) \right]^{-1}, \tag{8}$$

with $\beta = 5$ and $\lambda = 1.2$. The overall contact score is then defined as $Q(t) = \langle q_{ij}(t) \rangle$, averaged over all native contacts. To separate folded and unfolded ensembles, we determine a midpoint threshold $Q_{1/2}$ from the kernel density estimate of the 320 K MD reference distribution. Specifically, we locate the deepest minimum in the range 0.45–0.90; if no minimum is found, we set $Q_{1/2} = 0.70$. This same value is then reused to evaluate both 450 K MD and all generative trajectories.

Given this threshold, we compute the folded probability of each frame as

$$p_{\text{fold}}(t) = \left[ 1 + \exp\left( -2s \left[ Q(t) - Q_{1/2} \right] \right) \right]^{-1}, \qquad s = 10, \tag{9}$$

and report its ensemble average $\bar{p}_{\text{fold}}$. Finally, the folding free energy at temperature $T$ is defined as

$$\Delta G = -k_{\text{B}} T \ln \left[ \frac{\bar{p}_{\text{fold}}}{1 - \bar{p}_{\text{fold}}} \right]. \tag{10}$$

**MD Oracle performance** To quantify MD Oracle performance for 100 and 500 samples, we hold out one replica and compare it against a reference set containing the other four replicas. For 1000 samples, we hold out two replicas and use the remaining three as the reference.

## B.4 TRAINING DETAILS

We follow the hyper-parameters of Jing et al. (2024b) as closely as possible. Throughout all experiments we use an exponential moving average with decay 0.999, and AdamW (Loshchilov & Hutter, 2017) optimizer with a learning rate of $1 \times 10^{-4}$. We use an architecture described in Appendix A.1 with 5 transformer layers, 384-dimensional token embeddings, 16-head multi-head attention, and an IPA layers with 4 heads, 32-dimensional head size, and 8 query–key as well as 8 value points.

For the tetrapeptide dataset we recreate the setup of Jing et al. (2024b), sampling MDGEN-1000 from the released checkpoint and training our MARS-FM model for 1000 epochs and MDGEN-200 for 2500 epochs, both with a batch size of 8.

On MD-Cath dataset, MARS-FM is run for 1000 epochs with batch size 8 while sampling training pairs from two source clusters ($x_0$) and two corresponding destination clusters ($x_1$) with 12 sequences per cluster; MDGen-100 is trained for the same epoch budget with batch size 1, and MDGen-20 for 2500 epochs with batch size 4.

## B.5 SAMPLING DETAILS

All trajectories are sampled from the first MD frame in the test set; for the MD-Cath data, we use the first frame of the first replica. MDGen is then applied autoregressively, where the last of generated frames at each rollout serves as the conditional input for the next rollout. For MDGen in parallel mode, all calls are conditioned on the initial MD frame; we simply call it multiple times independently.

For MARS-FM ⊕ MDGen, we first draw a set of $\frac{\text{total number of frames}}{\text{MDGen window size}}$ frames using MARS-FM in parallel. Each output of MARS-FM is then used as the conditional frame for an independent MDGen rollout.

For hierarchical MARS-FM sampling, we begin from the initial frame and sample 200 first-layer children in parallel. Each of these nodes is then expanded once, and the process continues recursively until the required number of frames is reached.

## B.6 COMPUTE RESOURCES

All preprocessing tasks (including TICA projection, Radius of Gyration calculation, Secondary Structures Fractions calculation, clustering), and evaluation metric computation (such as divergence measures, RMSD, and RMSF), are performed on CPU nodes of the compute cluster.

Training and sampling are conducted on a single NVIDIA H100 GPU. Training on the tetrapeptide dataset takes approximately 1–2 days, while training on the MD-Cath dataset requires around 4–6 days. For sampling, generating 10,000 conformations for the full tetrapeptides test set takes approximately 1.5 hours; for MD-CATH, generating 500 conformations for all test domains takes around 5 hours.

## B.7 BASELINE CAVEATS

**BioEmu** For larger systems, we also benchmark against BioEmu (Lewis et al., 2024). Several caveats should be noted when interpreting these results. First, BioEmu uses a backbone-only protein representation, whereas MARS-FM explicitly models both backbone and side-chain torsion angles. Second, BioEmu was trained on substantially more data, including proprietary MD simulations, which makes direct comparisons less controlled. Third, BioEmu currently provides only an inference pipeline, with no publicly available training code. Despite these limitations, we performed inference evaluations on MD-Cath using the pre-trained BioEmu model, generating 500 conformations for comparison with reference MD simulations.

**Other Methods** Several other works have aimed, similar to MDGen (Jing et al., 2024b), at speeding up molecular dynamics, but they exhibit important limitations. The Implicit Transfer Operator (ITO)

(Schreiner et al., 2023) relies on a coarse-grained C$\alpha$ representation and assumes a fully connected graph with $\mathcal{O}(M^2)$ scaling, making application to systems with thousands of atoms infeasible. TimeWarp (Klein et al., 2023), based on RealNVP and a Transformer stack, incurs prohibitive training costs even for short peptides and has only been demonstrated on tetrapeptides. EquiJump (Costa et al., 2024) does not provide publicly available code and has so far only reported performance on the same fast-folding proteins used for training, without evidence of generalization.

## C  ADDITIONAL RESULTS

### C.1  SUPPLEMENTARY TETRAPEPTIDES TICA PLOTS

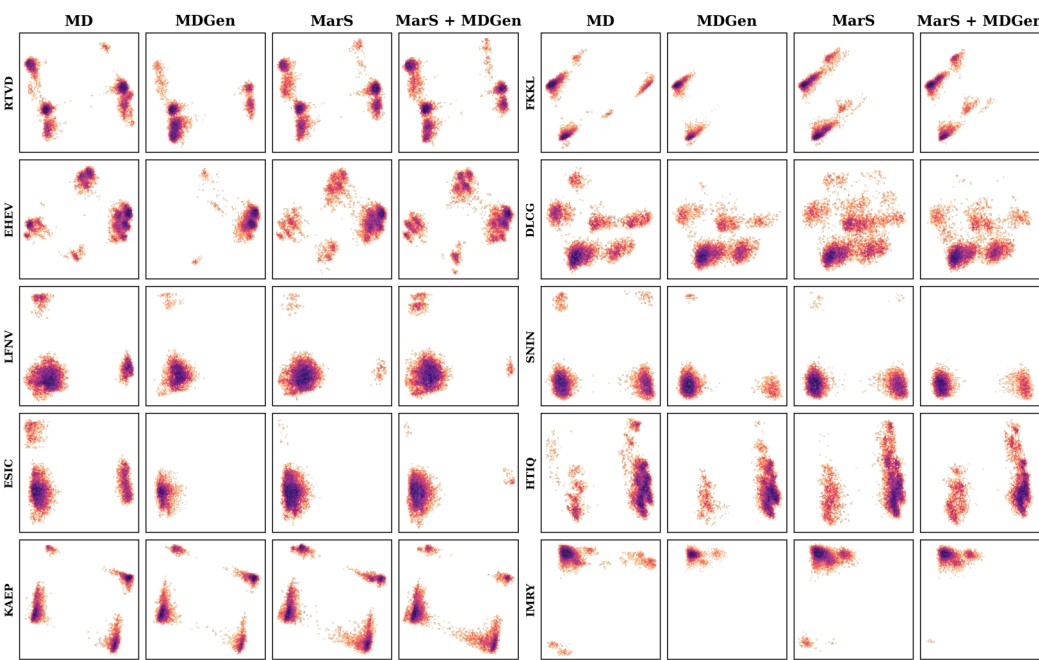

Figure 6: TICA plot for 10 random peptides in the test set, comparing MD ground-truth, MDGen, MARS (ours) and MARS + MDGen (ours). Our frameworks explore modes that are otherwise entirely ignored by MDGen.

### C.2  ADDITIONAL DYNAMICAL OBSERVABLES FOR TETRAPEPTIDES

To complement the thermodynamic metrics in the main text, we report additional dynamical observables on the 4AA tetrapeptide benchmark. Following MDGen (Jing et al., 2024b), we consider the normalized torsion autocorrelation

$$C_\theta(\Delta t) = \big\langle \cos\big(\theta_t - \theta_{t+\Delta t}\big)\big\rangle,$$

and define a relaxation time $\tau_\theta$ as the lag at which $C_\theta(\Delta t)$ decays below $1/e$ (up to 10 ns).

Figure 8 compares backbone and sidechain decorrelation curves for representative test peptides (ALDA, IWHF, KSIY, PGKM, SSNN) across MDGen-1000, MARS-FM, and the hybrid MARS-FM $\oplus$ MDGen-200 scheme. Figure 7 correlates model and MD relaxation times for all backbone and sidechain torsions in the test set.

Across all three models, MDGen-1000 unsurprisingly provides the closest match to MD torsion relaxation times, as it is explicitly trained to emulate fixed-lag MD transitions at 10 ps. In contrast, MARS-FM alone shows systematically shorter relaxation times and faster decay of $C_\theta(\Delta t)$, reflecting its design as an MSM-Emu: it is trained to sample across MSM states rather than to reproduce fine-grained torsional kinetics. The hybrid MARS-FM $\oplus$MDGen-200 interpolates between these behaviors, recovering near-MD agreement in $\tau_\theta$ while retaining the improved exploration of metastable states and macrostate free energies highlighted in Tables 1 and 2.

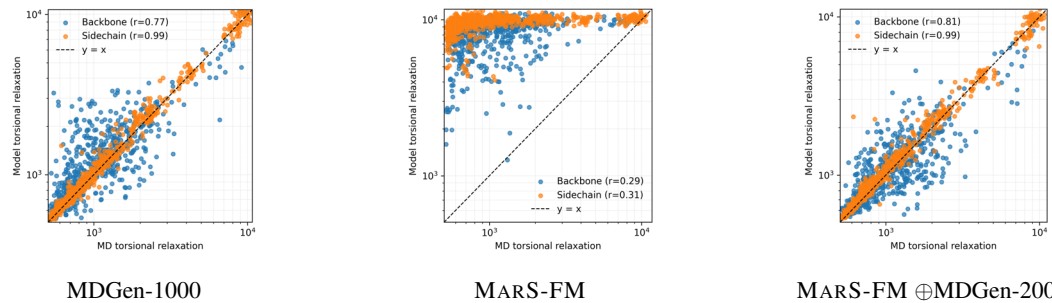

MDGen-1000                    MARS-FM                    MARS-FM ⊕MDGen-200

Figure 7: **Torsion relaxation times for 4AA tetrapeptides.** Scatter plots of model versus MD relaxation times $\tau_\theta$ for backbone (blue) and sidechain (orange) torsions. Axes are logarithmic; the dashed line indicates $y = x$. Pearson correlations $r$ for backbone and sidechain torsions are reported in the legend of each panel.

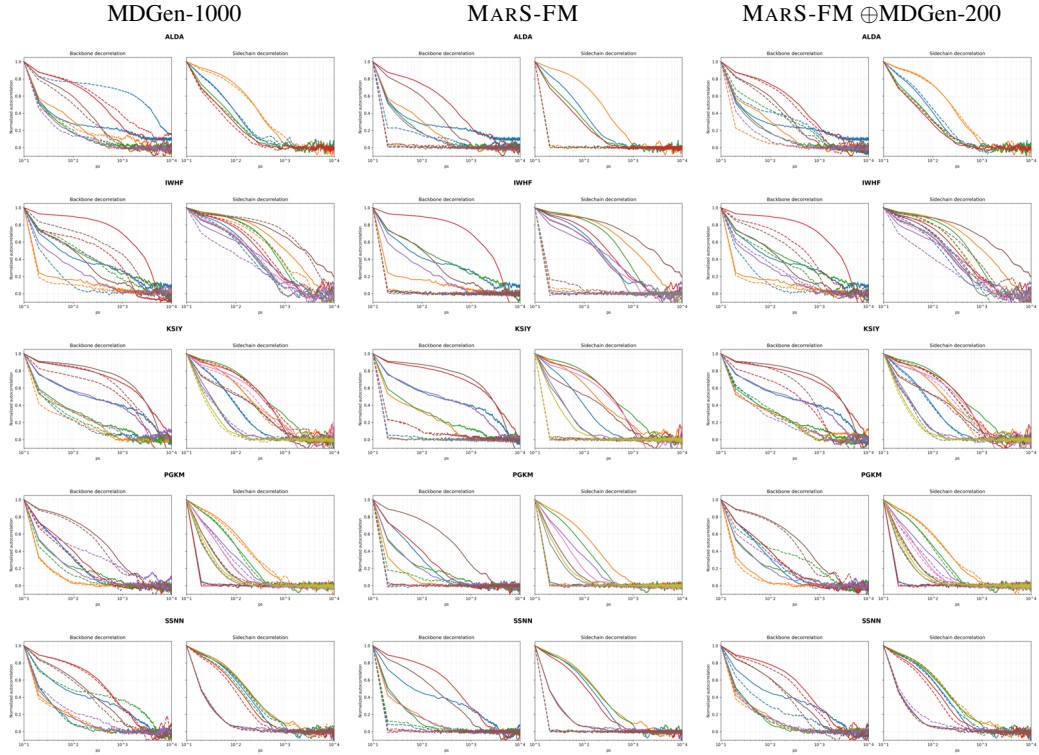

Figure 8: **Backbone and sidechain torsion decorrelation for 4AA tetrapeptides.** Normalized autocorrelation $C_\theta(\Delta t)$ as a function of lag time for backbone (left panels in each image) and sidechain (right panels) torsions in five representative test peptides (rows: ALDA, IWHF, KSIY, PGKM, SSNN) and three models (columns: MDGen-1000, MARS-FM, MARS-FM ⊕MDGen-200). Solid lines denote MD reference trajectories, dashed lines the corresponding model trajectories.

## C.3 Supplementary plots of generated samples from MD-Cath domains

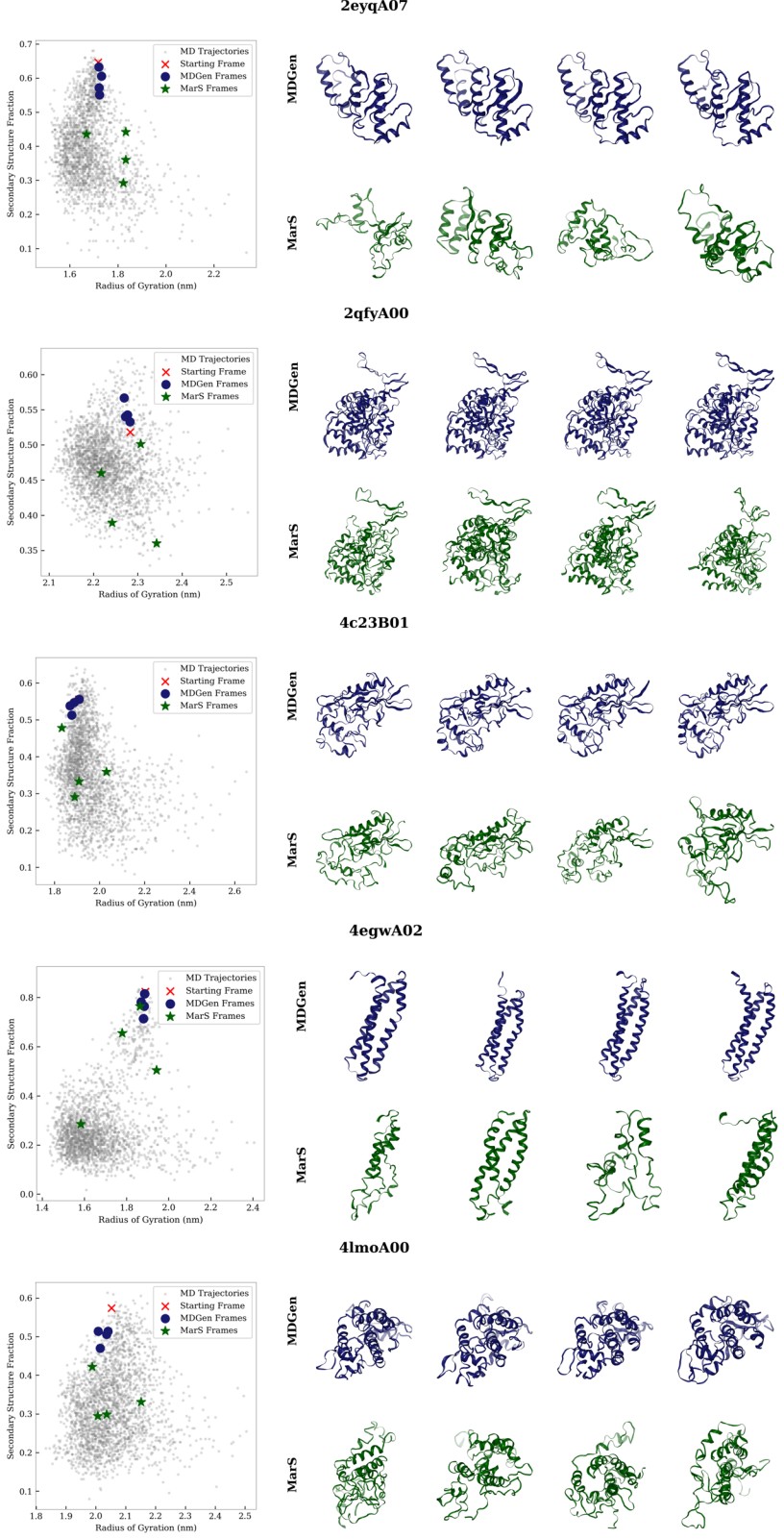

Figure 9: First 4 samples generated by MDGen and MARS-FM for 5 random domains in the MD-Cath test set.

## C.4 Ablation on hierarchical tree sampling

MARS-FM samples MD-Cath trajectories using hierarchical tree sampling (§5.2), where each node spawns $N$ children in parallel before the tree is further expanded. To study the impact of this branching factor, we fix the total sample budget to 500 conformations per domain on the 450 K MD-Cath test set (§5.2) and vary $N \in \{10, 20, 50, 100, 250, 500\}$. We then evaluate the same structural and thermodynamic metrics as in Table 2.

Results are reported in Table 5. Very small trees ($N = 10$) require more expansion depth to reach the target sample budget, increasing the number of autoregressive steps and leading to noticeable degradation in structural flexibility and folding free energies. Increasing $N$ to intermediate values ($N \approx 100$–$250$) consistently improves RMSD/RMSF correlations and reduces divergences in radius of gyration and secondary-structure distributions. However, pushing $N$ too high ($N = 500$) makes the tree very shallow: most samples are generated close to the initial frame, which harms state-space exploration and MSM reconstruction, as reflected by the higher MSM JSD and weaker RMSF correlations. Overall, intermediate branching factors around $N = 100$–$250$ provide the best trade-off between limiting temporal error accumulation and accurately recovering the MSM and long-timescale observables; our default choice of $N = 200$ used in the main experiments lies in this regime.

Table 5: Effect of branching factor $N$ in hierarchical MARS-FM tree sampling on the MD-Cath 450 K test set. We report Pearson $r \uparrow$ for RMSD and RMSF, forward KL divergence $\downarrow$, JSD $\downarrow$ of gyration radius, secondary-structure fractions, and MSM distributions, and folding free energy MAE (kcal/mol) $\downarrow$. Results are based on 500 generated conformations per domain and averaged over 5 inference runs.

| | Pairwise RMSD $r$ | Global RMSF $r$ | Per target RMSF $r$ | Gyration Radius KL | Gyration Radius JSD | Secondary Structure KL | Secondary Structure JSD | MSM JSD | $\Delta G_{\text{fold}}$ MAE |
|---|---|---|---|---|---|---|---|---|---|
| $N = 10$ | $0.54 \pm .007$ | $0.64 \pm .005$ | $0.89 \pm .002$ | $0.56 \pm .001$ | $0.13 \pm .001$ | $2.12 \pm .018$ | $0.24 \pm .001$ | $0.17 \pm .002$ | $2.38 \pm .005$ |
| $N = 20$ | $0.62 \pm .005$ | $0.69 \pm .005$ | $0.90 \pm .002$ | $0.51 \pm .002$ | $0.11 \pm .000$ | $1.54 \pm .003$ | $0.18 \pm .001$ | $0.18 \pm .007$ | $1.73 \pm .010$ |
| $N = 50$ | $0.65 \pm .001$ | $0.71 \pm .001$ | $0.90 \pm .002$ | $0.52 \pm .003$ | $0.10 \pm .000$ | $1.31 \pm .007$ | $0.17 \pm .000$ | $0.17 \pm .003$ | $1.43 \pm .019$ |
| $N = 100$ | $0.65 \pm .002$ | $0.71 \pm .002$ | $0.90 \pm .003$ | $0.52 \pm .002$ | $0.10 \pm .001$ | $1.13 \pm .007$ | $0.15 \pm .001$ | $0.18 \pm .002$ | $1.23 \pm .016$ |
| $N = 250$ | $0.65 \pm .003$ | $0.71 \pm .002$ | $0.89 \pm .002$ | $0.57 \pm .003$ | $0.10 \pm .001$ | $0.87 \pm .007$ | $0.12 \pm .001$ | $0.19 \pm .005$ | $1.03 \pm .003$ |
| $N = 500$ | $0.61 \pm .001$ | $0.69 \pm .002$ | $0.84 \pm .003$ | $0.87 \pm .003$ | $0.12 \pm .000$ | $0.83 \pm .010$ | $0.12 \pm .001$ | $0.25 \pm .004$ | $1.20 \pm .004$ |

## C.5 Ablation on MSM state construction for MD-Cath 450 K

In the main MD-Cath experiments (§5.2), we construct per-domain MSMs by $k$-means clustering of normalized radius of gyration and secondary-structure fractions into $c = 10$ metastable states with lag time $l = 50$ ns. To assess the sensitivity of MARS-FM to this choice, we perform an ablation over the MSM discretization on the 450 K test set, varying both the number of clusters $c$ and the MSM lag time $l$ while keeping all other settings fixed.

Specifically, we retrain MARS-FM and the hybrid MARS-FM $\oplus$ MDGen-20 using MSMs built with $(c, l) \in \{(10, 50), (10, 100), (10, 20), (20, 50), (5, 50)\}$, where $c$ denotes the number of $k$-means clusters and $l$ the lag time (in ns) used to estimate the transition matrix. For each configuration we generate 500 conformations per test domain. Results are reported in Table 6.

Overall, we observe that MARS-FM is robust to reasonable changes in the MSM discretization. For both MARS-FM and MARS-FM $\oplus$ MDGen-20, all $(c, l)$ settings yield very similar RMSD/RMSF correlations and divergences on radius of gyration and secondary-structure distributions. Increasing the MSM lag time from 50 ns to 100 ns slightly improves MSM JSD and $\Delta G_{\text{fold}}$ MAE, while shortening the lag to 20 ns degrades the gyration-radius KL and MSM JSD for both models. Varying the number of clusters at fixed lag time has only modest effects: coarse discretizations with $c = 5$ tend to worsen secondary-structure KL/JSD and MSM JSD, whereas $c = 20$ slightly improves MSM recovery without changing the overall conclusions. Taken together, these results indicate that the default MSM choice ($c = 10, l = 50$ ns) used in the main text offers a good trade-off across metrics, and that our improvements over MD-Emus do not rely on a finely tuned state construction.

Table 6: Ablation on MSM discretization for MD-Cath 450 K. Pearson $r \uparrow$ for RMSD and RMSF, forward KL divergence $\downarrow$ and JSD $\downarrow$ of gyration radius, fraction of secondary structures, MSMs distribution, and folding free energy MAE (kcal/mol) $\downarrow$. Results based on sampled trajectories of 500 conformations compared to ground-truth distributions. MSMs are labeled by the number of clusters $c$ and lag time (e.g., $c = 10$, lag 50 ns).

| MSM (clusters, lag) | Model | Pairwise RMSD r | Global RMSF r | Per target RMSF r | Gyration Radius KL | Gyration Radius JSD | Secondary Structures KL | Secondary Structures JSD | MSM JSD | $\Delta G_{fold}$ MAE |
|---|---|---|---|---|---|---|---|---|---|---|
| $c = 10$, lag 50 ns | MARS-FM $\oplus$ MDGen-20 | 0.63 | 0.69 | 0.83 | 0.98 | 0.13 | 0.73 | 0.11 | 0.24 | 1.02 |
| | MARS-FM | 0.65 | 0.71 | 0.89 | 0.55 | 0.10 | 0.93 | 0.13 | 0.19 | 1.05 |
| $c = 10$, lag 100 ns | MARS-FM $\oplus$ MDGen-20 | 0.69 | 0.73 | 0.85 | 0.70 | 0.12 | 0.74 | 0.12 | 0.20 | 0.91 |
| | MARS-FM | 0.69 | 0.73 | 0.90 | 0.50 | 0.11 | 1.05 | 0.15 | 0.15 | 1.21 |
| $c = 10$, lag 20 ns | MARS-FM $\oplus$ MDGen-20 | 0.56 | 0.66 | 0.82 | 1.07 | 0.14 | 0.75 | 0.12 | 0.25 | 0.97 |
| | MARS-FM | 0.53 | 0.65 | 0.86 | 0.72 | 0.11 | 1.17 | 0.16 | 0.21 | 0.98 |
| $c = 20$, lag 50 ns | MARS-FM $\oplus$ MDGen-20 | 0.64 | 0.69 | 0.83 | 0.94 | 0.13 | 0.76 | 0.12 | 0.24 | 0.93 |
| | MARS-FM | 0.59 | 0.67 | 0.89 | 0.64 | 0.11 | 1.10 | 0.14 | 0.20 | 0.96 |
| $c = 5$, lag 50 ns | MARS-FM $\oplus$ MDGen-20 | 0.62 | 0.68 | 0.83 | 0.88 | 0.13 | 0.93 | 0.14 | 0.24 | 0.95 |
| | MARS-FM | 0.57 | 0.65 | 0.89 | 0.57 | 0.10 | 1.58 | 0.18 | 0.20 | 1.23 |

## C.6 JSD FOR PAIRWISE RMSD AND GLOBAL RMSF, AND PEARSON $r$ FOR FOLDING FREE ENERGIES

To further quantify structural flexibility and thermodynamic accuracy, we report the mean JSD on the test set between reference MD samples and 500 samples generated by our models and baselines, along with the Pearson correlation ($r$) for folding free energies ($\Delta G_{fold}$). The results are presented in Table 7. For each protein, trajectories are aligned with respect to the first MD frame of the first replica before calculating the metrics. Both hybrid sampling (MARS-FM $\oplus$ MDGen) and hierarchical sampling (MARS-FM) more effectively capture the structural distribution of the trajectories and recover folding free energy correlations than the extended MDGEN baselines, often by a large margin.

Table 7: JSD $\downarrow$ for pairwise RMSD and global RMSF, and Pearson $r \uparrow$ for Folding Free Energies ($\Delta G_{fold}$). Results based on sampled trajectories of 500 conformations compared to ground-truth distributions and averaged over 5 inference runs.

| | Pairwise RMSD JSD | Global RMSF JSD | $\Delta G_{fold}$ r |
|---|---|---|---|
| MD (Oracle) | $0.16 \pm .005$ | $0.09 \pm .003$ | $0.90 \pm .002$ |
| MDGen-100 | $0.22 \pm .008$ | $0.31 \pm .009$ | $0.82 \pm .006$ |
| MDGen-20 | $0.38 \pm .005$ | $0.52 \pm .010$ | $0.85 \pm .005$ |
| MDGen-100 (in parallel) | $0.54 \pm .004$ | $0.58 \pm .004$ | $0.83 \pm .004$ |
| MDGen-20 (in parallel) | $0.73 \pm .002$ | $0.71 \pm .001$ | $0.86 \pm .001$ |
| BioEmu | $0.70 \pm .001$ | $0.72 \pm .001$ | $0.71 \pm .002$ |
| MARS-FM $\oplus$ MDGen-20 | $\mathbf{0.15} \pm .001$ | $\mathbf{0.24} \pm .001$ | $\mathbf{0.88} \pm .001$ |
| MARS-FM | $\underline{0.18} \pm .001$ | $\underline{0.28} \pm .001$ | $\underline{0.87} \pm .003$ |

## C.7 VALIDITY METRICS

We report validity metrics that directly probe native-contact preservation, $C_\alpha$–$C_\alpha$ geometry, and backbone covalent structure on the MD-Cath 450 K test set. Following the EquiJump (Costa et al., 2024), we compute Jensen–Shannon divergences (JSDs) between MD and model-generated trajectories for fraction of native contacts (FNC) and for the Global Distance Test–Total Score (GDT-TS). We additionally report backbone bond- and angle-RMSZ values.

**Fraction of native contacts.** FNC quantifies how well a sampled conformation preserves the interatomic contact pattern of a reference structure (Best et al., 2013). Given the set of native contacts $\mathcal{C}_{ref}$ defined on the reference structure (pairs of atoms closer than a cutoff $d_{cut}$), the FNC of a conformation $x$ is

$$\text{FNC}(x) = \frac{\#\{(i,j) \in \mathcal{C}_{ref} : d_{ij}(x) \le d_{cut}\}}{\#\mathcal{C}_{ref}}, \tag{11}$$

and we report the JSD between the FNC distributions of generated and MD trajectories.

**Global Distance Test–Total Score.** GDT-TS measures $C_\alpha$-level similarity to a reference structure (Zemla, 2003). For a conformation $x$ with $C_\alpha$ coordinates $\{r_i(x)\}_{i=1}^L$ and reference coordinates

$\{r_i^*\}_{i=1}^L$, we calculate,

$$\text{GDT-TS}(x) = \frac{1}{4} \sum_{d \in \{1,2,4,8\}\,\text{Å}} \frac{\#\{i \in \{1,\ldots,L\} : \|r_i(x) - r_i^*\| \le d\}}{L}\,. \tag{12}$$

We again compute the JSD between the GDT-TS distributions of generated and MD trajectories.

**Backbone covalent-geometry RMSZ.** To assess bonding quality, we compute backbone bond-length and bond-angle root-mean-square $Z$-scores (RMSZ) (Chen et al., 2010). For each backbone bond $k$, we form a $Z$-score

$$z_k = \frac{\ell_k - \ell_k^{\text{ideal}}}{\sigma_k}\,, \tag{13}$$

where $\ell_k$ is the observed bond length, and $\ell_k^{\text{ideal}}$ and $\sigma_k$ are the Engh–Huber crystallographic target mean and standard deviation for that bond type (Engh & Huber, 1991). The bond RMSZ is then

$$\text{Bond RMSZ} = \sqrt{\frac{1}{N_{\text{bond}}} \sum_{k=1}^{N_{\text{bond}}} z_k^2}\,, \tag{14}$$

with an analogous definition for the angle RMSZ over backbone bond angles. Values close to the MD oracle indicate MD-like covalent geometry, whereas large RMSZ reflects systematic distortions.

Table 8 summarizes these additional validity metrics. Both MARS-FM and MARS-FM $\oplus$ MDGen-20 achieve FNC- and GDT-TS-based JSDs that are closest to MD, indicating that they best reproduce MD-like distributions of native contacts and $C_\alpha$ distances. For covalent geometry, the MDGen-20/100 parallel variants obtain the lowest RMSZ by construction, as they perform only minimal moves around a single conditioning structure; however, they underperform strongly on dynamical diversity metrics (Table 2). In contrast, MARS-FM models explore configuration space substantially more, yet still markedly improve bond and angle RMSZ over MDGen and BioEmu.

Table 8: JSD $\downarrow$ for fraction of native contacts (FNC) and GDT-TS, and RMSZ $\downarrow$ for backbone bond lengths and bond angles. Results are based on sampled trajectories of 500 conformations compared to ground-truth MD distributions and averaged over 5 inference runs.

|  | FNC JSD | GDT-TS JSD | Bond RMSZ | Angle RMSZ |
|---|---|---|---|---|
| MD (Oracle) | $0.13 \pm .003$ | $0.11 \pm .004$ | $2.23 \pm .012$ | $2.19 \pm .009$ |
| MDGen-100 | $0.20 \pm .004$ | $0.17 \pm .004$ | $76.58 \pm 2.041$ | $5.27 \pm .077$ |
| MDGen-20 | $0.30 \pm .003$ | $0.22 \pm .002$ | $407.82 \pm 5.857$ | $7.00 \pm .031$ |
| MDGen-100 (in parallel) | $0.29 \pm .001$ | $0.31 \pm .002$ | $3.87 \pm .009$ | $2.56 \pm .002$ |
| MDGen-20 (in parallel) | $0.40 \pm .001$ | $0.42 \pm .001$ | $2.88 \pm .001$ | $2.32 \pm .001$ |
| BioEmu | $0.53 \pm .001$ | $0.49 \pm .001$ | $15.25 \pm 3.151$ | $2.90 \pm .001$ |
| MARS-FM $\oplus$ MDGen-20 | $0.18 \pm .001$ | $0.16 \pm .000$ | $8.34 \pm .121$ | $2.68 \pm .001$ |
| MARS-FM | $0.15 \pm .001$ | $0.12 \pm .001$ | $12.62 \pm .144$ | $3.06 \pm .001$ |

### C.8 MD-CATH AT 320K

In contrast to the high-temperature replicas, trajectories at 320 K display no large-scale domain motions. Both the radius of gyration and secondary structure fractions exhibit nearly an order-of-magnitude lower variability (Table 9). Similarly, all-atom RMSF and secondary structure fluctuations remain tightly clustered around their native values. These statistics confirm that proteins largely remain stable at lower temperatures, with no significant conformational changes—consistent with the findings reported by Mirarchi et al. (2024).

For methodological consistency, we built the Markov State Model (MSM) with the same hyper-parameters at 320 K as for the 450 K regime (10 clusters and 50 ns lag time). However, MSM could be temperature-specific, for instance, by shortening lag times or changing the number of clusters to capture subtler motions. As shown in Tables 10 and Table 11, MARS-FM continues to outperform all baselines in reproducing structural flexibility, radius of gyration and MSM state reconstruction. All methods show only marginal differences on secondary-structure KL/JSD which could be potentially improved by hyper-tuning MSM settings.

Table 9: Structural variability at 320K versus 450K in the MD-CATH dataset: standard deviations of radius of gyration and secondary-structure fractions, plus mean pairwise RMSD and mean all-atom RMSF

|  | MD-Cath 320K | MD-Cath 450K |
|---|---|---|
| Radius of Gyration std (nm) | $0.045 \pm .058$ | $0.265 \pm .164$ |
| Secondary Structures Fractions std | $0.046 \pm .024$ | $0.104 \pm .040$ |
| Pairwise RMSD (Å) | $4.44 \pm 2.93$ | $14.54 \pm 4.71$ |
| All Atom RMSF (Å) | $2.81 \pm 2.10$ | $10.34 \pm 3.49$ |

Table 10: Pearson $r \uparrow$ for RMSD and RMSF, forward KL divergence $\downarrow$ and JSD $\downarrow$ of gyration radius, fraction of secondary structures, MSMs distribution, and folding free energy MAE (kcal/mol) $\downarrow$ **at 320K**. Results based on sampled trajectories of 500 conformations compared to ground-truth distributions and averaged over 5 inference runs.

|  | Pairwise RMSD r | Global RMSF r | Per target RMSF r | Gyration Radius KL | Gyration Radius JSD | Secondary Structures KL | Secondary Structures JSD | MSM JSD | $\Delta G_{\text{fold}}$ MAE |
|---|---|---|---|---|---|---|---|---|---|
| MD (Oracle) | $0.90 \pm .009$ | $0.88 \pm .005$ | $0.91 \pm .003$ | $0.82 \pm .028$ | $0.10 \pm .003$ | $0.37 \pm .017$ | $0.05 \pm .002$ | $0.13 \pm .007$ | $0.32 \pm .003$ |
| MDGen-100 | $0.82 \pm .011$ | $0.78 \pm .004$ | $0.81 \pm .007$ | $1.50 \pm .022$ | $0.20 \pm .003$ | $0.76 \pm .014$ | $0.13 \pm .001$ | $0.18 \pm .002$ | $0.77 \pm .009$ |
| MDGen-20 | $0.79 \pm .018$ | $0.75 \pm .010$ | $0.79 \pm .012$ | $1.30 \pm .065$ | $0.19 \pm .007$ | $0.78 \pm .009$ | $0.19 \pm .002$ | $0.19 \pm .002$ | $1.10 \pm .013$ |
| MDGen-100 (in parallel) | $0.87 \pm .004$ | $0.83 \pm .002$ | $0.87 \pm .003$ | $1.60 \pm .027$ | $0.18 \pm .002$ | $\mathbf{0.44} \pm .004$ | $\mathbf{0.07} \pm .001$ | $\underline{0.14} \pm .002$ | $0.64 \pm .012$ |
| MDGen-20 (in parallel) | $0.83 \pm .004$ | $0.81 \pm .002$ | $0.85 \pm .003$ | $2.10 \pm .018$ | $0.21 \pm .001$ | $0.71 \pm .005$ | $\underline{0.09} \pm .001$ | $0.18 \pm .002$ | $1.10 \pm .004$ |
| BioEmu | $0.58 \pm 001$ | $0.63 \pm 004$ | $0.84 \pm 002$ | $2.67 \pm 042$ | $0.36 \pm 001$ | $0.94 \pm 011$ | $0.15 \pm 001$ | $0.25 \pm 003$ | $0.83 \pm .003$ |
| MARS-FM ⊕ MDGen-20 | $\mathbf{0.91} \pm .001$ | $\mathbf{0.87} \pm .001$ | $\mathbf{0.89} \pm .001$ | $\underline{0.74} \pm .001$ | $\mathbf{0.13} \pm .001$ | $\underline{0.46} \pm .001$ | $\underline{0.09} \pm .001$ | $\mathbf{0.10} \pm .001$ | $\underline{0.62} \pm .001$ |
| MARS-FM | $\underline{0.90} \pm .001$ | $\mathbf{0.87} \pm .001$ | $\mathbf{0.90} \pm .003$ | $\mathbf{0.72} \pm .004$ | $\underline{0.14} \pm .001$ | $0.68 \pm .009$ | $0.12 \pm .001$ | $\underline{0.14} \pm .001$ | $\mathbf{0.58} \pm .001$ |

Table 11: Pearson $r \uparrow$ for RMSD and RMSF, forward KL divergence $\downarrow$ and JSD $\downarrow$ of gyration radius, fraction of secondary structures, MSMs distribution, and folding free energy MAE (kcal/mol) $\downarrow$ **at 320K**. Results based on sampled trajectories of 100 / 1000 conformations compared to ground-truth distributions and averaged over 5 inference runs.

|  | Pairwise RMSD r | Global RMSF r | Per target RMSF r | Gyration Radius KL | Gyration Radius JSD | Secondary Structures KL | Secondary Structures JSD | MSM JSD | $\Delta G_{\text{fold}}$ MAE |
|---|---|---|---|---|---|---|---|---|---|
| MD (Oracle) | 0.87 / 0.93 | 0.85 / 0.91 | 0.88 / 0.93 | 2.21 / 0.44 | 0.18 / 0.07 | 0.96 / 0.19 | 0.09 / 0.04 | 0.43 / 0.08 | 0.58 / 0.32 |
| MDGen-100 | 0.82 / 0.75 | 0.77 / 0.73 | 0.81 / 0.77 | 3.43 / 1.15 | 0.29 / 0.20 | 1.23 / 1.00 | $\underline{0.12}$ / 0.20 | 0.20 / 0.22 | 0.70 / 1.15 |
| MDGen-20 | 0.82 / 0.74 | 0.79 / 0.68 | 0.83 / 0.73 | 2.96 / 1.25 | 0.25 / 0.22 | $\underline{1.08}$ / 1.05 | $\mathbf{0.11}$ / 0.22 | 0.18 / 0.24 | 0.62 / 1.08 |
| MDGen-100 (in parallel) | 0.81 / $\underline{0.88}$ | 0.77 / 0.85 | 0.81 / $\underline{0.88}$ | 3.52 / 1.24 | 0.30 / $\mathbf{0.13}$ | 1.19 / $\mathbf{0.33}$ | $\mathbf{0.11}$ / $\mathbf{0.06}$ | 0.21 / $\underline{0.13}$ | 0.71 / $\underline{0.63}$ |
| MDGen-20 (in parallel) | 0.81 / 0.83 | 0.78 / 0.81 | 0.83 / 0.86 | 3.26 / 1.86 | 0.27 / 0.20 | 1.28 / 0.62 | $\underline{0.12}$ / $\underline{0.09}$ | 0.20 / 0.18 | 1.10 / 1.09 |
| BioEmu | 0.58 / 0.58 | 0.62 / 0.63 | 0.83 / 0.84 | 4.07 / 2.39 | 0.40 / 0.35 | 1.57 / 0.86 | 0.18 / 0.14 | 0.42 / 0.24 | 0.84 / 0.82 |
| MARS-FM ⊕ MDGen-20 | $\mathbf{0.89}$ / $\mathbf{0.90}$ | $\mathbf{0.86}$ / $\mathbf{0.87}$ | $\underline{0.87}$ / $\mathbf{0.90}$ | $\underline{2.23}$ / $\mathbf{0.57}$ | $\underline{0.22}$ / $\mathbf{0.12}$ | $\mathbf{0.89}$ / $\underline{0.40}$ | $\underline{0.12}$ / $\underline{0.09}$ | $\mathbf{0.14}$ / $\mathbf{0.11}$ | $\underline{0.60}$ / $\underline{0.63}$ |
| MARS-FM | $\mathbf{0.89}$ / $\mathbf{0.90}$ | $\mathbf{0.86}$ / $\mathbf{0.87}$ | $\mathbf{0.89}$ / $\mathbf{0.90}$ | $\mathbf{1.99}$ / $\underline{0.61}$ | $\mathbf{0.21}$ / $\underline{0.13}$ | 1.25 / 0.60 | 0.15 / 0.12 | $\underline{0.15}$ / $\underline{0.13}$ | $\mathbf{0.59}$ / $\mathbf{0.58}$ |

# D  LLM USAGE

We used large language models (GPT-5) to assist in improving the grammar of the manuscript, as well as facilitate writing code.

# E  BROADER IMPACTS

The MARS-FM model's ability to generate realistic protein conformations over two orders of magnitude faster than conventional MD can accelerate drug discovery and other socially beneficial molecular design tasks, while also reducing the energy footprint of large-scale simulations. Conversely, the same speed-ups and accessibility could facilitate malicious protein engineering or foster overconfidence when the model is applied outside its intended scope.

# F  REPRODUCIBILITY STATEMENT

All details necessary to reproduce the experiments are present in the paper. All code necessary to reproduce our results has been included in the supplementary material as part of the submission and will be released publicly upon acceptance.

