# OpenReview forum: "MarS-FM: Generative Modeling of Molecular Dynamics via Markov State Models"
_ICLR.cc/2026/Conference — ICLR 2026 Poster_

### Official Review · Reviewer_3Wf3 · 2025-10-19

**Soundness:** 3
**Presentation:** 3
**Contribution:** 3
**Rating:** 6
**Confidence:** 4

**Summary:**

This work presents a new class of generative models called MSM Emulators, which is designed to overcome the issue in previous approaches where learning transition densities with fixed time steps may cause the training signal to be dominated by frequent but uninformative transitions. During training, the proposed model, dubbed as MarS-FM, leverages flow matching to learn state transitions defined by the discrete transition matrix of preconstructed Markov State Models (MSMs) rather than those based on fixed time steps, thereby enhancing its ability to cross energy barriers and access different macroscopic states. Experiments on the tetrapeptide and MD-CATH datasets demonstrate that MarS-FM outperforms existing methods across all metrics by a substantial margin, while exhibiting a large improvement of sampling from rare metastable states.

**Strengths:**

- This paper presents clear motivation and well-structured logic. Aiming to address the limitation of existing methods that learn transition densities at fixed time steps, it proposes a generative model based on the Markov State Model (MSM) that directly learns state transitions from the probabilistic transition matrix, enabling the model to cross energy barriers and access different macroscopic states.
- Extensive evaluations on the tetrapeptide and MD-CATH datasets have been conducted. The results demonstrate the model’s out-of-distribution (OOD) generalization to unseen biomolecular systems and its substantially superior ability to access diverse metastable states compared with existing methods.
- The training objective is simple and effective, making the method easy to reproduce. Remarkably, even with a single objective, the model achieves impressive experimental results, demonstrating the effectiveness of the proposed approach.

**Weaknesses:**

- As a comparison to existing models that learn transition densities at fixed time steps, MarS-FM appears to lack an analysis of dynamical observables for the generated trajectories, such as the decorrelation time of torsion angles. Evaluating the model solely using thermodynamic metrics does not allow one to distinguish it from an unconditional generative model that targets the Boltzmann ensemble, and therefore fails to highlight the advantages of the conditional generation approach.

**Questions:**

1. Equation (5) presents the flow-matching-based training objective, but I would like to know more about certain training details. For two macroscopic states $i$ and $j$ with corresponding samples $x(t)$ and $x_1$, the conformational difference between them may be substantial, making the conditioning on $x(t)$ potentially uninformative for generating $x_1$. Would this increase the training difficulty, and does the conditioning network indeed provide effective guidance in such cases?
2. Lines 264–278 describe two generation strategies built on MarS-FM. I would like to know whether, and under what conditions, each strategy can unbiasedly reproduce the Boltzmann distribution, which is crucial for MD simulations.
3. The Tree sampling strategy mentions the number of samples $n$ generated in parallel at each iteration. I would like to understand how the choice of $n$ affects the resulting ensemble under a fixed sampling budget. Intuitively, a larger $n$ may concentrate generated conformations within local states and miss global diversity, whereas a smaller $n$ may explore macroscopic states more broadly but lack local detail. The authors are encouraged to provide an ablation study on $n$ and investigate whether there exists a hyperparameter setting that achieves an optimal trade-off between local accuracy and global coverage.
4. As mentioned in the Weaknesses section, the authors could introduce dynamical metrics for further evaluation to verify whether MarS-FM can capture the true dynamical behavior of the MD process.

**Details Of Ethics Concerns:**

None.

---

> ### Author Response · Authors · 2025-11-23
> **Response to Reviewer 3Wf3 (1/2)**
>
> We would like to thank the reviewer for their thoughtful and constructive review of our work. We are pleased that the reviewer found that our paper has *clear motivation and well-structured logic*, and appreciated that our method **directly learns state transitions from the MSM transition matrix to cross energy barriers and access diverse macroscopic states**. We also appreciate the recognition of our *extensive evaluation on tetrapeptide and MD-CATH with strong OOD generalization and substantially improved metastable-state coverage*. We now provide responses to the main points raised by the reviewer.
>
> ---
>
> ## Weaknesses
>
> > ***”As a comparison to existing models that learn transition densities at fixed time steps, MarS-FM appears to lack an analysis of dynamical observables for the generated trajectories, such as the decorrelation time of torsion angles. Evaluating the model solely using thermodynamic metrics does not allow one to distinguish it from an unconditional generative model that targets the Boltzmann ensemble, and therefore fails to highlight the advantages of the conditional generation approach.”***
>
> We appreciate this suggestion and have added dynamical evaluations in the revised manuscript. In App. C.2 we now report backbone and side-chain torsion-angle relaxation statistics following the MDGen protocol. Across all three models, MDGen-1000 unsurprisingly provides the closest match to MD torsion relaxation times, as it is explicitly trained to emulate fixed-lag MD transitions at 10 ps. In contrast, MarS-FM alone shows systematically shorter relaxation times and faster decay of $C_\theta(\Delta t)$, reflecting its design as an MSM-Emu: it is trained to efficiently sample across MSM states rather than to reproduce fine-grained torsional kinetics at a given physical lag. The hybrid MarS-FM$\oplus$MDGen-200 interpolates between these behaviours, recovering near-MD agreement in $\tau_\theta$ while retaining the improved exploration of metastable states and macrostate free energies highlighted in our main MD-CATH tables.
> MarS-FM by itself prioritises metastable exploration and equilibrium MSM statistics over exact torsion kinetics, while the hybrid MarS-FM$\oplus$MDGen scheme restores MD-like torsion relaxation without sacrificing the gains in rare-state coverage and thermodynamic accuracy.
>
> ---
>
> ## Questions
>
> > ***1. Equation (5) presents the flow-matching-based training objective, but I would like to know more about certain training details. For two macroscopic states $i$ and $j$  with corresponding samples $x(t)$ and $x_1$, the conformational difference between them may be substantial, making the conditioning on $x(t)$  potentially uninformative for generating $x_j$. Would this increase the training difficulty, and does the conditioning network indeed provide effective guidance in such cases?***
>
> Thank you for this question. We agree that transitions between distant macrostates $i$ and $j$ can be structurally large, and this is precisely the challenging regime MarS-FM is designed to address.
>
> In our implementation, the model predicts **residual roto–translational updates** relative to $x(t)$, rather than generating $x_1$ from scratch in absolute coordinates. This makes the conditioning on $x(t)$ structurally meaningful: the flow always starts from the current geometry and applies incremental, state-dependent updates, so the network cannot simply ignore $x(t)$ and behave like an unconditional generator.
>
> Regarding the concern that $x(t)$ might be “uninformative” for a distant target state $j$: each MSM macrostate aggregates fragments from many replicas and multiple revisits of the same basin. For a given transition $i \to j$, MarS-FM is therefore trained on *many* heterogeneous pairs $(x_t, x_1)$ spanning different realisations of this transition. This is a key advantage over fixed-lag training on $x_t \to x_{t+\Delta t}$, where most pairs correspond to small, local moves dominated by fast vibrations. In our MSM-based setup, the conditioning network learns how typical configurations in state $i$ are mapped into typical configurations in state $j$ across all observed paths, which **increases transition diversity and exposure to barrier-crossing moves** rather than making the conditioning useless.
>
> We agree that such large moves are intrinsically harder to learn, and this relates to the autoregressive error accumulation we discuss in the paper. However, in practice training remains stable, and the added difficulty is outweighed by the gains in metastable-state coverage and diversity reported in our experiments.

---

> > ### Author Response · Authors · 2025-11-23
> > **Response to Reviewer 3Wf3 (2/2)**
> >
> > > ***2. Lines 264–278 describe two generation strategies built on MarS-FM. I would like to know whether, and under what conditions, each strategy can unbiasedly reproduce the Boltzmann distribution, which is crucial for MD simulations.***
> >
> > We thank the reviewer for raising this important point. In practice, neither of our strategies can be truly “unbiased” w.r.t. the *exact* Boltzmann distribution, because we only ever see finite MD data. Instead, both are designed to reproduce the **equilibrium behaviour encoded in MD**.
> >
> > Concretely, MarS-FM is trained to match the equilibrium of the MSM fitted to MD trajectories. In the infinite-data/ideal-model limit, MarS-FM sampling would therefore recover the **MSM equilibrium ensemble**—a coarse-grained approximation to the Boltzmann distribution defined by the MD used to build the MSM, not the unknown ground-truth Boltzmann distribution itself.
> >
> > The hybrid MarS-FM$\oplus$MDGen strategy combines this MSM-based exploration with a fixed-lag MD emulator. MDGen is trained directly on $x_t \to x_{t+\Delta t}$ transitions, while MarS-FM proposes state-resolved moves along the MSM graph. If both components were perfect emulators of the underlying MD process (MDGen at the chosen lag and MarS-FM for the MSM), then the hybrid sampler would also converge to the same MD-induced equilibrium ensemble. In reality, both models are approximate, so we do not claim exact Boltzmann sampling; instead, we quantify how close we are to MD through equilibrium observables such as folding free energies ($\Delta G_{\text{fold}}$) between MD and model-generated trajectories. In these metrics, the hybrid MarS-FM$\oplus$MDGen strategy yields the closest match to MD in our experiments (Table 2 in the manuscript).
> >
> > ---
> >
> > > ***3. The Tree sampling strategy mentions the number of samples  generated in parallel at each iteration. I would like to understand how the choice of n affects the resulting ensemble under a fixed sampling budget. Intuitively, a larger may concentrate generated conformations within local states and miss global diversity, whereas a smaller n may explore macroscopic states more broadly but lack local detail. The authors are encouraged to provide an ablation study on n and investigate whether there exists a hyperparameter setting that achieves an optimal trade-off between local accuracy and global coverage.***
> >
> > We thank the reviewer for this suggestion. In the revised manuscript we have added a dedicated **ablation over the branching factor \(n\)** in Tree Sampling (Appendix C4). The table below reports the effect of $n$ on MD-CATH 450K metrics:
> >
> > | \(n\) | Pairwise RMSD \(r\) | Global RMSF \(r\) | Per-target RMSF \(r\) | Gyration Radius KL | Gyration Radius JSD | Secondary Structure KL | Secondary Structure JSD | MSM JSD | \(\Delta G_{\text{fold}}\) MAE |
> > |-:|-|-|-|-|-|-|-|---------|-------------------------------|
> > | 10   | 0.54 | 0.64 | 0.89 | 0.56 | 0.13 | 2.12 | 0.24 | 0.17 | 2.38 |
> > | 20   | 0.62 | 0.69 | 0.90 | 0.51 | 0.11 | 1.54 | 0.18 | 0.18 | 1.73 |
> > | 50   | 0.65 | 0.71 | 0.90 | 0.52 | 0.10 | 1.31 | 0.17 | 0.17 | 1.43 |
> > | 100  | 0.65 | 0.71 | 0.90 | 0.52 | 0.10 | 1.13 | 0.15 | 0.18 | 1.23 |
> > | 250  | 0.65 | 0.71 | 0.89 | 0.57 | 0.10 | 0.87 | 0.12 | 0.19 | 1.03 |
> > | 500  | 0.61 | 0.69 | 0.84 | 0.87 | 0.12 | 0.83 | 0.12 | 0.25 | 1.20 |
> >
> >
> > We observe that very small $n$ (e.g., 10) leads to worse structural correlations and much poorer thermodynamic/MSM metrics; with small $n$, most of the budget is spent on deep rollouts along relatively few branches, accumulating more model error over time. Very large $n$ (e.g., 500) improves some local ensemble metrics (e.g., secondary-structure KL) but harms others such as gyration-radius KL and MSM JSD, consistent with the reviewer’s intuition that excessively large $n$ leads to local over-exploration within a limited set of states and reduced global balance. Intermediate values (around $n = 50$–$250$) provide the best overall trade-off: good structural correlations, low KL/JSD for gyration radius and secondary structure, and the lowest $\Delta G_{\text{fold}}$ MAE at $n = 250$. We now included this ablation and discussion in the updated manuscript (Appendix C4).
> >
> > ---
> >
> > > ***4. As mentioned in the Weaknesses section, the authors could introduce dynamical metrics for further evaluation to verify whether MarS-FM can capture the true dynamical behavior of the MD process.***
> >
> > Please see the response in the Weaknesses section above.
> >
> > ---
> >
> > We thank the reviewer again for their valuable feedback and thoughtful questions, which led us to include new dynamical analyses and Tree Sampling ablations that have strengthened the paper. We hope that our rebuttal addresses the concerns—particularly regarding **dynamical observables** and the **effect of the number of branches in Tree Sampling**—and we kindly ask the reviewer to consider upgrading their score if satisfied with our responses. We would be very happy to clarify any remaining questions.

---

> ### Comment · Reviewer_3Wf3 · 2025-11-24
>
> Thank you for the authors' response. I believe that the explanations and additional experiments have resolved the vast majority of my concerns. In particular, the ablation study on the Tree Sampling method effectively clarifies the capability boundaries of the proposed approach and improves its interpretability. Overall, I consider this to be a good piece of work, and I will raise my score accordingly.

---

> > ### Author Response · Authors · 2025-11-28
> >
> > Thank you very much for your positive follow-up. We are glad the additional explanations and Tree Sampling ablation resolved your concerns. We sincerely appreciate your decision to raise the score to 8.

---

### Official Review · Reviewer_2MiT · 2025-10-21

**Soundness:** 3
**Presentation:** 2
**Contribution:** 2
**Rating:** 6
**Confidence:** 3

**Summary:**

This paper introduces a novel strategy to improve modeling of molecular dynamics (MD) based on the introduction of a Markov State Model to better represent state transitions at training time. The paper introduces MarS-FM as a model in this space based on Flow-Matching. MarS-FM  sampled molecular trajectories are compared against samples from MDGen and traditional MD simulations to demonstrate that the proposed method can achieve superior fidelity, even when unfolding simulations of molecules not observed during training.

**Strengths:**

* The paper tackles a practical problem in molecular modeling, demonstrating the effectiveness of the proposed MarS-FM solution against a relevant method in literature.

* The paper reports convincing results on a large set of peptides demonstrating the generalization capabilities of MarS-FM on a disjoint set of trajectories using a wide range of metrics.

* The paper is overall well-motivated and effectively explains limitation of existing MD Emulators with clear illustrations.

**Weaknesses:**

* The key part of the method and its training objective is quite compressed in a few paragraphs in section 4.1. Without any additional pseudocode, illustrations, and references, the details are quite hard to follow.
   * The notation for the noise sample $x_0$, initial $x(t)$, next $x_1$ and interpolated $x_s$ frames is quite confusing. The time index $t$ is dropped in $x_1$ (and $x_s$), which should depend on $t$ and $\tau$.

   * The paper reports that $x_1$ is sampled uniformly withing the state $S_j$. In practice, I expect that the density $p(x_1|S_j)$ is far from uniform on its domain. Please clarify the distinction between the empirical and modeled density in this paragraph.

    * A methodology comparison with MDGen is mentioned but not fully detailed. The method and background sections describe MSM constructions and input representations but the notation and flow-matching objective are hardly introduced, which makes following the objective in equation 5 more challenging.

* The paper reports extensive empirical comparisons with MDGen but limited or no ablation on the effect of the modeling choices such as the lag time $\tau$ or number of MSM states. The reported metrics focus on statistics of the equilibrium distribution, but no comparison or ablation evaluates the fidelity of the transitions learned by MarS-FM.

**Questions:**

1. Section 4.1 reports that the MSM construction is performed once per dataset. Does this mean that different molecules share the same state space or that only the TICA projection are performed on the whole dataset but the clustering and transition modeling are molecule-specific? In the first case, how can a small state space (e.g. 10 x 10 transitions) effectively capture the dynamics of multiple molecules?


2. Can the authors clarify the sampling strategy described in lines 248-251? The paper reports that $x_1$ is sampled uniformly withing the state $S_j$. In practice, the density $p(x_1|S_j)$ is far from uniform on its domain. Do the authors sample uniformly from the data-points that are assigned to $S_j$ in the training trajectory?

 3. How are the intervals $\tau$ picked in practice at training time? The parameter $\tau$ is implicit in $p_T(x_1|x(t))$ but it seems quite a relevant detail to balance the focus on different temporal horizons. Furthermore, this parameter may affect  the accuracy of the MSM model since a limited number of states (e.g. 10) may not be sufficient to describe smaller lag times since the transitions. How does Mars-FM work for varying $\tau$ and number of MSM states?

 4. How does the transition distribution modeled by Mars-FM compare against MDGen or other models in literature that model a specific transition interval $\tau$? Do the conditional statistic also improve or does Mars-FM improve only in terms of the statistics of the equilibrium distribution?

---

> ### Author Response · Authors · 2025-11-23
> **Response to Reviewer 2MiT(1/3)**
>
> We thank the reviewer for their thoughtful and constructive evaluation of our work. We are encouraged that they found our study to *“tackle a practical problem in molecular modeling”*. We also appreciate the recognition that the paper *“reports convincing results on a large set of peptides”*, highlighting the strong generalization of MarS-FM to a disjoint collection of trajectories and its performance across a broad suite of metrics. Finally, we are grateful for the positive assessment that the paper is *“overall well-motivated”* and *“effectively explains limitation of existing MD emulators with clear illustrations.”* Below, we respond point-by-point to the reviewer’s comments and provide additional analyses and clarifications.
>
> ---
>
> ## Weaknesses
>
> ###  1. Clarity
> We thank the reviewer for these helpful suggestions. In the updated manuscript, all new or revised text is highlighted in blue.
>
> > “The key part of the method and its training objective is quite compressed in a few paragraphs in section 4.1. Without any additional pseudocode, illustrations, and references, the details are quite hard to follow.”
>
> We expanded the **Training objective** part of Sec. 4.1 (around Eq. (5)) to spell out how $x_0, x(t), x_1$, and $x_s$ are sampled and used in Eq. (5), and we now explicitly point to Appendix A.1–A.2 (p.16–17) for pseudocode and the interpolation/velocity formula.
>
> > “The notation for the noise sample $x_0$, initial $x(t)$, next $x_1$ and interpolated $x_s$ frames is quite confusing. The time index \(t\) is dropped in $x_1$ (and $x_s$), which should depend on $t$ and $\tau$.”
>
> In the same paragraph (Sec. 4.1), we now explicitly distinguish the Gaussian noise sample $x_0$ from MD frames. We also state that $x_1 \sim p_T(\cdot\mid x(t))$ is an MD frame at lag time $\tau$ from $x(t)$, clarify that the $(t,\tau)$ dependence is implicit for readability, and explain how $x_s$ and $\dot x_s$ are constructed (with details in App. A.1–A.2).
>
> > “The paper reports that $x_1$ is sampled uniformly within the state $S_j$. In practice, I expect that the density $p(x_1\mid S_j)$ is far from uniform on its domain. Please clarify the distinction between the empirical and modeled density in this paragraph.”
>
> We now clarify in Sec. 4.1 that “uniform within a state” means uniformly over the **finite set of MD frames** assigned to $S_j$ by the MSM. Appendix A.2 (p.17) spells out the three-step procedure for drawing training pairs, making explicit that this is an empirical approximation to $p_T(\cdot\mid x(t))$.
>
> > “A methodology comparison with MDGen is mentioned but not fully detailed. The method and background sections describe MSM constructions and input representations but the notation and flow-matching objective are hardly introduced, which makes following the objective in equation 5 more challenging.”
>
> We made the relationship to MDGen explicit in Sec. 4.1: the **Representation** paragraph now states that MarS-FM uses the same SE(3) representation and MDGen-style velocity network, and the paragraph immediately after Eq. (5) explains that, unlike MDGen’s fixed-lag transitions $x_t \rightarrow x_{t+N}$, MarS-FM samples $x_1$ via the MSM transition kernel $T$, providing a more diverse, state-conditioned set of training pairs while preserving the MSM-implied timescale.

---

> > ### Author Response · Authors · 2025-11-23
> > **Response to Reviewer 2MiT(2/3)**
> >
> > ###  2. Ablation
> > > ***The paper reports extensive empirical comparisons with MDGen but limited or no ablation on the effect of the modeling choices such as the lag time  or number of MSM states. The reported metrics focus on statistics of the equilibrium distribution, but no comparison or ablation evaluates the fidelity of the transitions learned by MarS-FM.***
> >
> >
> > Thank you for this suggestion. In response, we have added a new ablation on MSM state construction and lag time for the MD-Cath 450 K benchmark (Appendix C5 of the updated manuscript). The numerical results for the MSM ablation are summarized in the table below.
> >
> > | MSM (c, lag)      | Model               | RMSD r | Global RMSF r | Per-target RMSF r | Rg KL | Rg JSD | Sec. frac. KL | Sec. frac. JSD | MSM JSD | ΔG_fold MAE |
> > |-------------------|---------------------|--------|----------------|-------------------|-------|--------|---------------|----------------|---------|-------------|
> > | c = 10, 50 ns     | MarS-FM + MDGen-20  | 0.63   | 0.69           | 0.83              | 0.98  | 0.13   | 0.73          | 0.11           | 0.24    | 1.02        |
> > | c = 10, 50 ns     | MarS-FM             | 0.65   | 0.71           | 0.89              | 0.55  | 0.10   | 0.93          | 0.13           | 0.19    | 1.05        |
> > | c = 10, 100 ns    | MarS-FM + MDGen-20  | 0.69   | 0.73           | 0.85              | 0.70  | 0.12   | 0.74          | 0.12           | 0.20    | 0.91        |
> > | c = 10, 100 ns    | MarS-FM             | 0.69   | 0.73           | 0.90              | 0.50  | 0.11   | 1.05          | 0.15           | 0.15    | 1.21        |
> > | c = 10, 20 ns     | MarS-FM + MDGen-20  | 0.56   | 0.66           | 0.82              | 1.07  | 0.14   | 0.75          | 0.12           | 0.25    | 0.97        |
> > | c = 10, 20 ns     | MarS-FM             | 0.53   | 0.65           | 0.86              | 0.72  | 0.11   | 1.17          | 0.16           | 0.21    | 0.98        |
> > | c = 20, 50 ns     | MarS-FM + MDGen-20  | 0.64   | 0.69           | 0.83              | 0.94  | 0.13   | 0.76          | 0.12           | 0.24    | 0.93        |
> > | c = 20, 50 ns     | MarS-FM             | 0.59   | 0.67           | 0.89              | 0.64  | 0.11   | 1.10          | 0.14           | 0.20    | 0.96        |
> > | c = 5, 50 ns      | MarS-FM + MDGen-20  | 0.62   | 0.68           | 0.83              | 0.88  | 0.13   | 0.93          | 0.14           | 0.24    | 0.95        |
> > | c = 5, 50 ns      | MarS-FM             | 0.57   | 0.65           | 0.89              | 0.57  | 0.10   | 1.58          | 0.18           | 0.20    | 1.23        |
> >
> > To directly address your concern, we recompute all MD-Cath 450 K metrics for both MarS-FM and MarS-FM + MDGen-20 while varying the MSM discretization: number of clusters $c \in \{5,10,20\}$ and lag time $\ell \in \{20,50,100\}\,\mathrm{ns}$. Overall, we observe that MarS-FM is robust to reasonable changes in the MSM discretization. For both MarS-FM and MarS-FM + MDGen-20, all $(c,\ell)$ settings yield very similar RMSD/RMSF correlations and divergences on radius of gyration and secondary-structure distributions, with only marginal fluctuations across the grid; importantly, all these variants remain substantially better than the MDGen and BioEmu baselines reported in Table 2. Taken together, these results indicate that the default MSM choice $(c=10,\ell=50~\mathrm{ns})$ used in the main text offers a good trade-off across metrics, **suggesting that our conclusions do not rely on a finely tuned discretization**.
> >
> > Additionally, following Reviewer 3Wf3’s comment, Appendix C4 now includes an ablation over the number of branches in the Tree sampling procedure used by MarS-FM. This experiment shows that our main advantages over MDGen persist across a reasonable range of branching factors, further **supporting that the reported gains are not tied to a single, brittle choice of modeling hyperparameters.**

---

> > > ### Author Response · Authors · 2025-11-23
> > > **Response to Reviewer 2MiT(3/3)**
> > >
> > > ## Questions
> > >
> > > ---
> > >
> > > > **1. Section 4.1 reports that the MSM construction is performed once per dataset. Does this mean that different molecules share the same state space … ?**
> > >
> > > We thank the reviewer for raising this point. We have clarified in the revised manuscript that MSM construction is performed once as an offline pre-processing step **to fix the MSM hyperparameters per dataset** (choice of observables, number of clusters, lag time), but we then build a **separate MSM for every training molecule/domain** from its own MD trajectories. This is now stated explicitly in Sec. 4.1, “Constructing MSMs”, where we also note that these per-molecule state spaces and transition matrices remain fixed during training and are used only to define training pairs. In addition, we now emphasize that **no MSM information (states or transition probabilities) is provided to the model at inference time**, so there is no leakage from training MSMs to test proteins.
> > >
> > > ---
> > >
> > >
> > > > **2. Can the authors clarify the sampling strategy described in lines 248–251? The paper reports that $x_1$ is sampled uniformly within the state $S_j$. In practice, the density $p(x_1 \mid S_j)$ is far from uniform on its domain. Do the authors sample uniformly from the data-points that are assigned to $S_j$ in the training trajectory?**
> > >
> > > We thank the reviewer for pointing this out. Yes, we sample **uniformly from the MD frames assigned to $S_j$** in the training trajectories, i.e., from the *discrete set* of data points belonging to that state, not from a continuous uniform density over the state region.
> > >
> > > To make this explicit, we have revised Sec. 4.1 (paragraph describing the sampling of $x_1$) to state that “uniform within a state” means uniformly over the finite set of MD frames assigned to $S_j$ by the MSM clustering, and we describe this again in Appendix A.2 when presenting the procedure for drawing training pairs.
> > >
> > > ---
> > >
> > > > **3. How are the intervals $\tau$ picked in practice at training time? … How does MarS-FM work for varying $\tau$ and number of MSM states?**
> > >
> > > We thank the reviewer for this question. As discussed in our response to **Weakness 2**, we now provide an explicit ablation over both the number of MSM states $c$ and lag time $\ell$ in Appendix C.5. These results (see also our response to Weakness 2) indicate that our conclusions do not rely on a finely tuned choice of lag time or number of MSM states, and that MarS-FM is robust to reasonable variations in $(c,\ell)$.
> > >
> > >
> > > ---
> > >
> > > > **4. How does the transition distribution modeled by MarS-FM compare against MDGen or other models in literature that model a specific transition interval $\tau$? Do the conditional statistics also improve or does MarS-FM improve only in terms of the statistics of the equilibrium distribution?**
> > >
> > > We thank the reviewer for this question. Conceptually, MarS-FM is not designed to learn the microscopic fixed-lag kernel $p(x(t+\tau)\mid x(t))$ in the same sense as MD-Emus such as MDGen. Instead, we first build per-protein MSMs and then train MarS-FM to sample from the *coarse-grained* MSM transition distribution $p_T(x_1 \mid x(t))$ between metastable states at a chosen lag time. Accordingly, our evaluation focuses on (i) equilibrium and structural statistics (RMSD/RMSF, $R_g$, secondary-structure fractions, $\Delta G_{\text{fold}}$) and (ii) the quality of the *MSM* transition distribution via MSM JSD. On MD-Cath, MarS-FM and MarS-FM+MDGen-20 consistently achieve lower MSM JSD than MDGen/BioEmu while also improving equilibrium metrics, indicating that conditional dynamics at the metastable level are improved, not just the stationary distribution. As suggested by Reviewer 3Wf3, we have additionally included dynamical observables in Appendix C.2 (torsion autocorrelations and relaxation times) to further illustrate how MarS-FM and the hybrid MarS-FM+MDGen behave compared to MD and MDGen at shorter time scales. If this still does not fully address the reviewer’s concern about “transition distributions,” we would be very happy to further clarify the intended comparison in a follow-up discussion.
> > >
> > >
> > > ---
> > >
> > > ## Summary
> > >
> > > We thank the reviewer again for their valuable feedback and thoughtful questions, which prompted us to add new ablations and clarifications to the manuscript. These additions indicate that MarS-FM is **robust to choices of MSM lag time, number of states, and sampling hyperparameters**. We hope that our rebuttal and manuscript revisions address the concerns—particularly regarding the training methodology —and we would be grateful if the reviewer could consider upgrading their score if they find the changes satisfactory. We would of course be very happy to clarify any remaining questions or requested revisions.

---

> ### Comment · Reviewer_2MiT · 2025-11-25
>
> I wish to thank the authors for the detailed answers and additional results.
> I do believe that the additional provided information and experimental evidence has improved the quality of the overall submission, hence I am increasing my score.

---

> > ### Author Response · Authors · 2025-11-28
> >
> > Thank you very much for your detailed follow-up. We are glad the additional information and experiments improved your assessment of the work, and we sincerely appreciate your decision to raise the score to 8.

---

### Official Review · Reviewer_H61Z · 2025-10-31

**Soundness:** 2
**Presentation:** 1
**Contribution:** 1
**Rating:** 2
**Confidence:** 5

**Summary:**

The authors developed a model called MarS-FM for combining molecular dynamics (MD) modeling using flow matching and a traditional method markov state model. The work is build upon MDGen, and using similar dataset to test the performance of the generated MD.

**Strengths:**

There are not many work in machine learning conference that combines the MSM with MD generation models. This MarS-FM model sets an good example for combining traditional model and recent developments.

**Weaknesses:**

The benchmark is only on short trajectories, which highly limit the ability of MSM model (which desiged for long trajectories) and the application of this model.

**Questions:**

1. Authors should test their models on long MD trajectories including DESRES fast folding proteins and dynamic proteins (including WW domain and BPTI). Using short trajectories from MDGen is not convincing for either MSM models and conventional simulations
2. This model should tested on large proteins compared with models like AlphaFold subsampling and BioEmu. Only small protein is limited to toy datasets
3. Authors should also report multiple quality metrics including the protein validity, the bonding, the interaction distances, the Ca-Ca distances, etc.
4. MSM models are the best fit for generate protein folding trajectories and pathway between rare event. I suggest authors add several case studies on these examples.
5. A large issue for the tICA representation of conformation generation models are that incorrect conformation and unfold conformation are mixed in the same place. Authors should give multiple examples of the generated conformations based on k-means clustering, to show the possible generated incorrect conformations.

---

> ### Author Response · Authors · 2025-11-23
> **Response to Reviewer H61Z (1/3)**
>
> We thank the reviewer for taking the time to evaluate our work and for highlighting that *“MarS-FM sets a good example for combining traditional MSMs with recent generative MD models”*. We now provide responses to the questions raised by the reviewer.
>
> ## Weaknesses
> > ***The benchmark is only on short trajectories, which highly limit the ability of MSM model (which designed for long trajectories) and the application of this model.***
>
> Our main benchmark is **MD-Cath**, not the short MDGen tetrapeptide trajectories. As described in Sec. 5.2 and App. B.1, MD-Cath contains thousands of protein domains (up to **500 residues**) with trajectories of **≈500 ns per replica**, multiple replicas per temperature. MSMs in MarS-FM are built on these MD-Cath trajectories, and all main results (Tables 2–4, App. Tables) are reported on this benchmark.
> If by “long trajectories” the reviewer has in mind a different timescale, we would be grateful if they could kindly indicate which public dataset and observables they would consider more appropriate for our stated goal of matching equilibrium ensembles — noting that the DESRES dataset suggested by the reviewer has trajectories only up to 100 ns.
>
>
> ## Questions
> > ***1. Authors should test their models on long MD trajectories including DESRES fast folding proteins and dynamic proteins [...]. Using short trajectories from MDGen is not convincing for either MSM models and conventional simulations.***
>
> Again, we do not restrict our evaluation to short MDGen trajectories. Our main evaluation and MSM construction are on **MD-Cath**[1] (Sec. 5.2, Tables 2–4, App. B.1–B.3), which provides hundreds of nanoseconds of MD per test protein (≈500 ns per replica, with 5 replicas at each temperature evaluated, with unfolding and large rearrangements).
> By contrast, **DESRES fast-folding proteins are typically ≤80 residues and trajectories are ≤100 ns**, i.e., about **5× shorter** than the 500 ns MD-Cath trajectories we use. DESRES is valuable but focuses on a small set of mini-proteins with limited sequence and fold diversity, whereas MD-Cath is designed for diverse, non-homologous proteins up to 500 residues and is therefore more aligned with our goal of assessing ensemble generation across many unseen proteins.
> We kindly invite the reviewer to re-read the respective parts of our paper.
>
> > ***2. This model should tested on large proteins compared with models like AlphaFold subsampling and BioEmu. Only small protein is limited to toy datasets***
>
> MD-Cath domains range **up to 500 residues**, with many in the 200–500 range [1]. We therefore do not consider this a “toy” dataset or one with “only small proteins”. In fact, **DESRES fast-folding systems are ≤80 residues**, i.e., smaller than most of the proteins we evaluate on MD-Cath.
> **If the reviewer believes that MD-Cath is a toy dataset, we would politely ask to provide a justification for such a statement**, and suggest which other open dataset would not be a “toy” one in their opinion.
> We draw the reviewer’s attention that we **already** compare directly against **BioEmu** on MD-Cath (Tables 2–3 main text; App. Tables 7, 8, 10, 11) and show that MarS-FM (and MarS-FM ⊕ MDGen) substantially outperform it on ensemble flexibility, structural distributions, MSM state recovery, and folding free energies. Moreover, BioEmu is **backbone-only**, whereas MarS-FM explicitly models both backbone and side-chain torsions.

---

> > ### Author Response · Authors · 2025-11-23
> > **Response to Reviewer H61Z (2/3)**
> >
> > > ***3. Authors should also report multiple quality metrics including the protein validity, the bonding, the interaction distances, the Ca-Ca distances, etc.***
> >
> > We appreciate this suggestion. In the revised manuscript we have added validity metrics that address this point (App. C7  “VALIDITY METRICS”, Tab. 8). Following EquiJump [2], we report the Jensen–Shannon divergence (JSD) between MD and model distributions of the fraction of native contacts (FNC) and the Global Distance Test–Total Score (GDT-TS) versus the native structure, which directly probe interaction distances and Cα–Cα–level similarity along the trajectory. In addition, we report standard backbone covalent-geometry metrics (bond and angle RMSZ), which quantify bonding quality [3].
> >
> > On FNC and GDT-TS, **MarS** and **MarS-FM ⊕ MDGen** are the best-performing generative models: their JSDs are the closest to MD. This shows that our method captures MD-like distributions of native contacts and Cα distances, rather than merely producing static native-like structures.
> >
> > For covalent geometry, MDGen-20/100 parallel achieve the lowest bond/angle RMSZ, because they run in short windows and therefore only make minimal moves around a single conditioning structure. However, these models perform massively worse on dynamical diversity metrics (e.g., KL of radius of gyration and secondary-structure content; see Tables 2 and 3 in the original manuscript), indicating severe under-exploration. In contrast, MarS and MarS-FM ⊕ MDGen explore configuration space substantially more, yet still markedly improve bond/angle RMSZ over MDGen and BioEmu.
> >
> > | | FNC JSD | GDT-TS JSD | bond RMSZ | angle RMSZ |
> > |-|-|-|-|-|
> > | MD (oracle) | 0.13| 0.11| 2.23| 2.19|
> > | MDGen-100| 0.20| 0.17| 76.58 | 5.27 |
> > | MDGen-20| 0.30| 0.22 | 407.82| 7.00|
> > | MDGen-100 parallel | 0.29| 0.31| 3.87| 2.56|
> > | MDGen-20 parallel| 0.40 | 0.42 | 2.88| 2.32|
> > | BioEmu| 0.53 | 0.49|15.25| 2.90|
> > | MarS-FM⊕MDGen| 0.18| 0.16| 8.34| 2.68|
> > | MarS-FM  | 0.15| 0.12 | 12.62| 3.06|
> >
> >
> >
> >
> >
> > > ***4. MSM models are the best fit for generate protein folding trajectories and pathway between rare event. I suggest authors add several case studies on these examples.***
> >
> > Within this equilibrium-focused scope, we already provide a thermodynamic validation via **folding free energies** $\Delta G_{\text{fold}}$ on MD-Cath, following the BioEmu protocol. We report both **MAE** (Table 2) and **Pearson r** (App. C.6, Table 7), and show that MarS-FM and MarS-FM ⊕ MDGen substantially improve $\Delta G_{\text{fold}}$ accuracy over MDGen variants and BioEmu, achieving correlations close to the MD oracle. Together with our ensemble metrics (RMSD/RMSF correlations, radius of gyration and secondary-structure distributions, MSM stationary distributions), this supports that MarS-FM captures the structural variability and thermodynamics of the MD ensemble.
> >
> > Extending MSM-Emus to explicit rare-event pathway reconstructions on dedicated ultra-long-trajectory datasets is an exciting direction, but lies beyond the scope of the present submission and the MD-Cath setting, which is designed for large-scale equilibrium ensembles across many proteins.
> >
> >
> >
> >
> > > ***5. A large issue for the tICA representation of conformation generation models are that incorrect conformation and unfold conformation are mixed in the same place. Authors should give multiple examples of the generated conformations based on k-means clustering, to show the possible generated incorrect conformations.***
> >
> > We would appreciate additional clarification on this comment. If we understand the concern correctly, the reviewer is worried that, in a tICA embedding, physically incorrect and unfolded but physically valid conformations may overlap, and that this could hide nonphysical generated structures.
> > We would first like to clarify that MarS-FM **does not generate trajectories in tICA space**. For tetrapeptides (Sec. 5.1), tICA is used only to obtain a low-dimensional representation in which we cluster frames into metastable states for MSM construction. For MD-Cath (Sec. 5.2, App. B.2), the MSMs that define the transition structure for MarS-FM are instead built without tICA, by clustering directly in normalized radius of gyration and secondary-structure fractions.
> > Structural correctness in our work is assessed in full 3D geometry using ensemble metrics: RMSD/RMSF correlations, distributions of radius of gyration and secondary-structure fractions, MSM stationary distributions, folding free energies, and (in the revised version) FNC/GDT-TS and bond/angle RMSZ. Across these observables, MarS-FM matches MD substantially better than MDGen and BioEmu.

---

> > > ### Author Response · Authors · 2025-11-23
> > > **Response to Reviewer H61Z (3/3)**
> > >
> > > We thank the reviewer for their comments. We hope that (1) our clarification of the role and timescales of the main MD-Cath benchmark (**proteins up to 500 residues with multiple trajectories of 500 ns**), (2) the explicit BioEmu comparison on larger proteins— **already contained in the original manuscript** — and (3) the newly added validity and distance-based metrics (FNC, GDT-TS, bond/angle RMSZ) help to address their concerns. If the reviewer finds these clarifications satisfactory, we would kindly ask them to increase their score, and we are happy to answer any further detailed questions.
> > >
> > > ---
> > > [1] Mirarchi, Antonio, Toni Giorgino, and Gianni De Fabritiis. "mdCATH: A large-scale MD dataset for data-driven computational biophysics."
> > >
> > > [2] Costa, Allan Dos Santos, et al. "Equijump: Protein dynamics simulation via so (3)-equivariant stochastic interpolants."
> > >
> > > [3] Chen, Vincent B., et al. "MolProbity: all-atom structure validation for macromolecular crystallography."

---

### Official Review · Reviewer_pGe2 · 2025-11-02

**Soundness:** 3
**Presentation:** 1
**Contribution:** 3
**Rating:** 4
**Confidence:** 4

**Summary:**

The paper proposes to use Markov state models to augment the training of generative models that sample molecular dynamics (MD-Emus) by targeting the distribution of transitions as defined by the MSM transition matrix. Such models are called MSM-Emus. The method is validated to yield comparable or improved results on tetrapeptide simulations and improved results on the mdCATH dataset of protein simulations.

**Strengths:**

The paper poses an insightful question for training MD rollout emulators - how do we more effectively learn from MD trajectory datasets beyond simply training on observed frame pairs? The approach is sound and seems to lead to useful empirical results.

**Weaknesses:**

**Clarity**

My chief concern is the clarity of the writing. First, the authors write that they "learn to sample from the Markov chain transition induced by an MSM, rather than emulating the noisy temporal dynamics induced by MD." This phrasing seems to suggest that the the MSM transitions somehow differ and improve upon, by design, the dynamics in the MD. In truth, the MSMs are themselves fit to the "noisy temporal dynamics induced by MD" by enforcing coarse-graining and Markovianity approximations. Second, MSMs are not usually taken to be representations of fine-grained dynamics, yet the present model still learns full structural dynamics. Third, the authors argue that the time lag is a limitation of MD-Emus, yet MSM-Emus also have a specific lag time - that used in the construction of the MSM.

A better way to express the proposed idea is as follows: Instead of learning the transition density faithfully, MSM-Emus are trained to predict next frames as a mixture distribution corresponding to MSM states in proportion to the MSM transition probabilities. While this corrupts the underlying dynamics, it allows the model to focus on learning the state change probabilities rather than paying attention (and potentially overfitting) to the exact starting frame within the state and for rare starting states to be identified and upsampled during training. This ultimately and empirically leads to better emulation of long-timescale dynamics (in the absence of the MSM) at inference time.

**Justification for score**
I would support to accept this paper if the authors can convincingly tidy up the presentation.

**Questions:**

No specific questions.

---

> ### Author Response · Authors · 2025-11-23
> **Rebuttal**
>
> We would like to thank the reviewer for their thoughtful review of our work and for helping us improve the clarity of the manuscript. We are grateful for stating that it *“poses an insightful question”* and that the *“approach is sound”* leading to *“useful empirical results”*. As your main concern pertains to the story and presentation, we address it directly below and list the concrete manuscript changes we have made in response to your feedback (highlighted in blue in the revised PDF).
>
> 1. **Clarifying the “noisy temporal dynamics” phrasing.**
>    We agree that the original wording *“rather than emulating the noisy temporal dynamics induced by MD”* was potentially misleading. We have removed this and now describe MSM-Emus as models that *“learn to sample from the Markov-chain transition induced by an MSM constructed from MD trajectories”* which *“provides a coarse-grained Markov model of long-timescale dynamics between metastable states”* (Sec. 1, *Main Contributions*).
>
> 2. **Explicitly acknowledging the trade-off in fine-grained dynamics.**
>    To address your concern that MSM-based training may corrupt the underlying dynamics, we now explicitly state in Sec. 4 that MSM-Emus are designed to preserve coarse-grained kinetics (metastable-state populations and transition probabilities), but are *“not intended to faithfully reproduce all fine-grained dynamical observables at the same level of accuracy as fixed-lag MD-Emus.”* We emphasize that training on state-to-state jumps *“prioritizes long-timescale behavior over the detailed intra-state time ordering”* and that MSM-Emus and MD-Emus are complementary. We point to Sec. 5 / App. C.2, where we report torsion-angle decorrelation curves and show how the hybrid MARS-FM ⊕ MDGen recovers near-MD torsional kinetics while retaining improved exploration.
>
> 3. **Adopting your “mixture over MSM states” description.**
>    In Sec. 4 (Definition 1), we have aligned the description of the class with your suggested phrasing. We now explicitly explain that, for $x(t) \in S_i$, the MSM defines a categorical distribution $j \mapsto T_{ij}$, and we *“interpret $p_T(\cdot \mid x(t))$ as the resulting mixture distribution over MD frames: one first samples a destination state $S_j$ according to $T_{ij}$ and then samples a conformation from the empirical ensemble of MD frames assigned to $S_j$. MSM-Emus are thus trained to draw conformations from this MSM-induced mixture, rather than to learn the exact frame-to-frame MD transition density.”*
>
> 4. **Clarifying the main novelty of MSM-Emus vs. fixed-lag MD-Emus.**
>    We have clarified that the key novelty is that the model is **not** trained solely on transitions of the form $x_t \rightarrow x_{t+N}$, for any fixed $N$. At the end of Sec. 3, we now emphasize that once trajectories have been clustered into metastable states, *“any pair of frames assigned to successive MSM states can be used for training, greatly increasing the number and diversity of informative transitions.”* In Sec. 4.1, we further specify that because each metastable state *“aggregates frames from different replicas and from multiple visits to the same basin within a trajectory,”* MARS-FM is exposed to transitions that *“span replicas and multiple barrier-crossing events, even when only a few such events occur in any individual time series,”* and that the effective training set is therefore *“no longer restricted to the specific $x(t) \rightarrow x(t+\tau)$ pairs obtained at a fixed frame-saving interval.”* Appendix A.2 also explicitly notes that our MSM-informed sampling procedure *“allows us to sample across different replicas.”*
>
> We thank the reviewer again for these helpful suggestions on clarifying the story, which we believe significantly improve the presentation. Given that clarity was the only weakness raised and we have now addressed it through targeted manuscript edits, we would be very happy to clarify any remaining concerns, and we hope the reviewer will consider increasing their score if satisfied with our responses.

---

### Author Response · Authors · 2025-11-23
**General Response to All Reviewers (1/2)**

We thank the reviewers for their thoughtful feedback and constructive questions that have helped us improve the submission significantly. We are glad to see that reviewers found our work *“reports convincing results on a large set of peptides demonstrating the generalization capabilities of MarS-FM on a disjoint set of trajectories using a wide range of metrics”* (R 2MiT), that *“The results demonstrate the model’s out-of-distribution (OOD) generalization to unseen biomolecular systems and its substantially superior ability to access diverse metastable states compared with existing methods”* (R 3Wf3), and that our framework *“poses an insightful question for training MD rollout emulators […] The approach is sound and seems to lead to useful empirical results”* (R pGe2).

In this general response, we address three points: (1) we provide an overview of the new experiments and ablations we ran to address questions raised in the reviews; (2) we summarize the main changes that clarify the manuscript; (3) We clarify some misunderstandings from one review regarding our experimental results. All new results and added/edited text are highlighted in blue in the updated manuscript.

---

### (1) New experiments and ablations

- **Dynamical observables for tetrapeptides (App. C.2).** (3Wf3)
We added backbone and side-chain torsion autocorrelation curves and relaxation times on the 4AA benchmark. These results show that the hybrid MarS-FM ⊕ MDGen-200 recovers near-MD torsion relaxation while retaining its improved metastable-state exploration and macrostate free energies.

- **Tree sampling branching-factor ablation (App. C.4, Tab. 5).**  (3Wf3)
We systematically varied the number of branches \(N\) in the hierarchical Tree Sampling scheme on MD-Cath 450 K. The results indicate that sampling is robust to this hyperparameter and that, across all tested values of \(N\), MarS-FM continues to outperform other models.

- **MSM discretization and lag-time ablation (App. C.5, Tab. 6).**  (2MiT)

  We ablated different MSM constructions on MD-Cath 450 K, varying the number of clusters and the lag time. Across this grid, metrics change only mildly, and all settings remain substantially better than MDGen/BioEmu. This confirms that our conclusions do not rely on a finely tuned MSM.

- **Additional structural validity metrics on MD-Cath (App. C.7, Tab. 8).**  (H61Z)

  Following EquiJump/BioEmu-style diagnostics, we added JSDs for fraction of native contacts (FNC) and GDT-TS, as well as backbone bond- and angle-RMSZ. MarS-FM and MarS-FM ⊕ MDGen achieve FNC/GDT-TS distributions closest to MD while still maintaining good covalent geometry, whereas MDGen variants that best preserve bonds/angles do so at the cost of severely under-exploring conformational space. The results support that MarS-FM improves both ensemble quality and validity, not just diversity.

---

### (2) Clarifications and presentation changes

- **Clearer positioning of MSM-Emus and their relationship to MD-Emus (Abstract; Secs. 1, 3–4).**

- **More precise description of the MSM-induced mixture transition distribution in the definition of MSM-Emus (Sec. 4, Definition 1).**

- **Sharper statement of the main novelty: training beyond fixed-lag frame pairs using transitions between metastable states (Secs. 3–4.1; App. A.2).**

- **More explicit relationship to MDGen (Sec. 4.1)**.

- **Explicit discussion of the trade-off between coarse-grained long-timescale behavior and fine-grained dynamics, including hybrid MSM-Emu ⊕ MD-Emu schemes (Sec. 4; Sec. 5; App. C.2).**

- **Expanded explanation of training objective and sampling procedure (Sec. 4.1, App. A.1–A.2).**

- **Clarification of MSM construction and per-protein state spaces (Sec. 4.1, App. B.2).**


All of these additions and edits are highlighted in blue in the revised PDF so that reviewers can easily verify the new experiments and the corresponding clarifications in the manuscript.

---

> ### Author Response · Authors · 2025-11-23
> **General Response to All Reviewers (2/2)**
>
> ---
>
> ### (3) Factual clarifications
> We noticed a few statements in the review from **Reviewer H61Z** that appear to stem from misunderstandings. **These are not the result of the rebuttal; they were already present and readily visible in the original submission’s main text**.
>
> - **Claim:** “Authors should test their models on long MD trajectories including DESRES.”
>   **Clarification:** Our main evaluation is on **MD-CATH** with trajectories **up to 500 ns**; DESRES fast-folding trajectories are ≤100 ns (i.e., ≥5× shorter than MD-CATH).
>
> - **Claim:** “Using short trajectories from MDGen is not convincing for either MSM models and conventional simulations.”
>   **Clarification:** We do not restrict evaluation to MDGen-style short trajectories. The **central results** are on **MD-CATH**, reported in Tables 2–4 (main text) and Tables 9–11 (appendix).
>
> - **Claim:** “Only small protein is limited to toy datasets.”
>   **Clarification:** **MD-CATH** includes proteins **up to 500 residues** (while DESRES is only up to ~80). MD-CATH is **not** a toy dataset; it is designed to test broad generalization across many proteins.
>
> - **Claim:** “Model should be tested on large proteins compared with models like AlphaFold subsampling and BioEmu.”
>   **Clarification:** We **already reported BioEmu** on **MD-CATH** in the **original submission** (see Tables 2–3 in the main text) and observed substantially better performance from MarS-FM.
>
> ---
> We once again thank all reviewers and the area chairs for their time and thoughtful evaluations. We address each reviewer’s specific questions and suggestions in separate, point-by-point responses accompanying this general reply.

---

### Meta-Review · Area_Chair_HnF1 · 2026-01-06

**Summary:**

Reviewer pGe2's concern is the clarity of the writing.

Reviewer H61Z' s concerns include: (1) The evaluation is limited to short MDGen trajectories, which fails to demonstrate the core strengths of MSMs in modeling long-timescale dynamics, folding pathways, and rare events; (2) The experiments are further restricted to small proteins, without meaningful comparisons on larger or more realistic systems or against strong baselines such as AlphaFold-based subsampling or BioEmu; (3) the evaluation lacks essential physical and structural quality metrics needed to assess the validity of generated conformations, and the paper does not include MSM-appropriate case studies (e.g., folding or transition pathways); (4) concerns remain about representation quality, particularly with tICA embeddings potentially mixing correct and incorrect conformations, which are not analyzed through clustering or qualitative visualization.

Reviewer 2MiT's main concerns are lack of clarity and ablation study.

Reviewer 3Wf3's concern is the lack of an analysis of dynamical observables for the generated trajectories.

**Reviewer Concerns:**

In response, the authors have implemented all of Reviewer pGe2’s concrete suggestions regarding clarity.

Reviewer H61Z: The rebuttal substantially addresses several of the reviewer’s core concerns by clarifying that the primary benchmark is MD-Cath rather than short MDGen trajectories, demonstrating that MSMs are constructed on hundreds of nanoseconds of MD data for proteins up to 500 residues, directly comparing against BioEmu on this benchmark, and adding a broad set of physically meaningful validity and distance-based metrics (e.g., FNC, GDT-TS, bond and angle RMSZ) that were previously missing. It also reasonably explains why explicit rare-event pathway case studies and ultra-long single-trajectory analyses fall outside the paper’s stated equilibrium-ensemble focus, and clarifies misconceptions about the role of tICA in the method. However, some concerns remain partially outstanding: the work still does not include explicit qualitative case studies of folding pathways or rare-event transitions, and the absence of experiments on canonical ultra-long folding datasets may continue to limit confidence for a reviewer who prioritizes dynamical pathway reconstruction over equilibrium ensemble matching.

Reviewer 2MiT and Reviewer 3Wf3's concerns have been fully addressed.

**Reviewer Scores:**

Reviewer pGe2 stated that he/she would support to accept this paper if the authors can convincingly tidy up the presentation. The rebuttal has addressed this concern, therefore Reviewer pGe2 would increase the rating to 6 or 8.

Given the rebuttal, Reviewer H61Z’s score would likely increase modestly but remain on the negative side. Several major criticisms, such as use of only short trajectories, reliance on toy proteins, lack of physical validity metrics, absence of BioEmu comparisons, and misunderstanding of tICA’s role, are convincingly addressed, which removes some factual objections that justified the very low initial score of 2. However, Reviewer H61Z’s core expectation of MSM-style rare-event and folding-pathway case studies on canonical long-trajectory datasets is not fully met and is instead deferred as out of scope.  Thus, a reasonable estimate is that the Reviewer H61Z would raise the score from 2 (reject) to 4 (marginally below the acceptance threshold).

Reviewer 2MiT is satisfied with the rebuttal and expresses a willingness to increase the rating to 8.

Reviewer 3Wf3 stated that the explanations and additional experiments have resolved the vast majority of his/her concerns. In particular, the ablation study on the Tree Sampling method effectively clarifies the capability boundaries of the proposed approach and improves its interpretability. Overall, he/she considered this paper to be a good piece of work, and would raise the score to 8.

The AC believes that the paper has adequately addressed the major concerns raised by the reviewers, and that most reviewers now express clear support for the work. The remaining concerns are minor in nature and have been reasonably explained and justified by the authors. On this basis, the AC finds the overall response satisfactory and recommends acceptance of the paper.

---

### Decision · Program_Chairs · 2026-01-26

Accept (Poster)